# Estimators for Multivariate Information Measures in General Probability Spaces

**Arman Rahimzamani**
Department of ECE
University of Washington
armanrz@uw.edu

**Himanshu Asnani**
Department of ECE
University of Washington
asnani@uw.edu

**Pramod Viswanath**
Department of ECE
University of Illinois at Urbana-Champaign
pramodv@illinois.edu

**Sreeram Kannan**
Department of ECE
University of Washington
ksreeram@uw.edu

## Abstract

Information theoretic quantities play an important role in various settings in machine learning, including causality testing, structure inference in graphical models, time-series problems, feature selection as well as in providing privacy guarantees. A key quantity of interest is the mutual information and generalizations thereof, including conditional mutual information, multivariate mutual information, total correlation and directed information. While the aforementioned information quantities are well defined in arbitrary probability spaces, existing estimators add or subtract entropies (we term them $\Sigma H$ methods). These methods work only in purely discrete space or purely continuous case since entropy (or differential entropy) is well defined only in that regime.

In this paper, we define a general graph divergence measure ($\mathbb{GDM}$), as a measure of incompatibility between the observed distribution and a given graphical model structure. This generalizes the aforementioned information measures and we construct a novel estimator via a coupling trick that directly estimates these multivariate information measures using the Radon-Nikodym derivative. These estimators are proven to be consistent in a general setting which includes several cases where the existing estimators fail, thus providing the only known estimators for the following settings: (1) the data has some discrete and some continuous valued components (2) some (or all) of the components themselves are discrete-continuous *mixtures* (3) the data is real-valued but does not have a joint density on the entire space, rather is supported on a low-dimensional manifold. We show that our proposed estimators significantly outperform known estimators on synthetic and real datasets.

## 1 Introduction

Information theoretic quantities, such as mutual information and its generalizations, play an important role in various settings in machine learning and statistical estimation and inference. Here we summarize briefly the role of some generalizations of mutual information in learning (cf. Sec. 2.1 for a mathematical definition of these quantities).

1. **Conditional mutual information** measures the amount of information between two variables $X$ and $Y$ given a third variable $Z$ and is zero iff $X$ is independent of $Y$ given $Z$. CMI finds a wide

range of applications in learning including causality testing [1, 2], structure inference in graphical models [3], feature selection [4] as well as in providing privacy guarantees [5].

2. **Total correlation** measures the degree to which a set of $N$ variables are independent of each other, and appears as a natural metric of interest in several machine learning problems, for example, in independent component analysis, the objective is to maximize the independence of the variables quantified through total correlation [6]. In feature selection, ensuring the independence of selected features is one goal, pursued using total correlation in [7, 8].

3. **Multivariate mutual information** measures the amount of information shared between multiple variables [9, 10] and is useful in feature selection [11, 12] and clustering [13].

4. **Directed information** measures the amount of information between two random processes [14,15] and is shown as the correct metric in identifying time-series graphical models [16–21].

Estimation of these information-theoretic quantities from observed samples is a non-trivial problem that needs to be solved in order to utilize these quantities in the aforementioned applications. While there has been long history in estimation of entropy [22–25], and renewed recent interest [26–28], much less effort has been spent on the multivariate versions. A standard approach to estimating general information theoretic quantities is to write them out as a sum or difference of entropy (denoted $H$ usually) terms which are then directly estimated; we term such a paradigm as $\Sigma H$ paradigm. However, the $\Sigma H$ paradigm is applicable only when the variables involved are all discrete or there is a joint density on the space of all variables (in which case, differential entropy $h$ can be utilized). The underlying information measures themselves are well defined in very general probability spaces, for example, involving mixtures of discrete and continuous variables; however, no known estimators exist.

We motivate the requirement of estimators in general probability spaces by some examples in contemporary machine learning and statistical inference.

1. It is common place in machine learning to have data-sets where **some variables are discrete, and some are continuous**. For example, in recent work on utilizing information bottleneck to understand deep learning [29], an important step is to quantify the mutual information between the training samples (which are discrete) and the layer output (which is continuous). The employed methodology was to quantize the continuous variables; this is common practice, even though highly sub-optimal.

2. Some variables involved in the calculation **may be mixtures of discrete and continuous variables**. For example, the output of ReLU neuron will not have a density even when the input data has a density. Instead, the neuron will have a discrete mass at $0$ (or wherever the ReLU breakpoint is) but will have a continuous distribution on the positive values. This is also the case in gene expression data, where a gene may have a discrete mass at expression $0$ due to an effect called drop-out [30] but have a continuous distribution elsewhere.

3. The variables involved may have a joint density **only on a low dimensional manifold**. For example, when calculating mutual information between input and output of a neural network, some of the neurons are deterministic functions of the input variables and hence they will have a joint density supported on a low-dimensional manifold rather than the entire space.

In the aforementioned cases, no existing estimators are known to work. It is not merely a matter of having provable guarantees either. When we plug in estimators that assume a joint density into data that does not, the estimated information measure can be strongly negative.

We summarize our main contributions below:

1. **General paradigm** (Section 2): We define a general paradigm of graph divergence measures which captures the aforementioned generalizations of mutual information as special cases. Given a directed acyclic graph (DAG) between $n$ variables, the graph divergence is defined as the Kullback-Leibler (KL) divergence between the true data distribution $\mathbb{P}_X$ and a restricted distribution $\overline{\mathbb{P}}_X$ defined on the Bayesian network and can be thought of as a measure of incompatibility with the given graphical model structure. These graph divergence measures are defined using the Radon Nikodym derivatives which are well-defined for general probability spaces.

2. **Novel estimators** (Section 3): We propose novel estimators for these graph divergence measures, which directly estimate the corresponding Radon-Nikodym derivatives. To the best of our knowl-

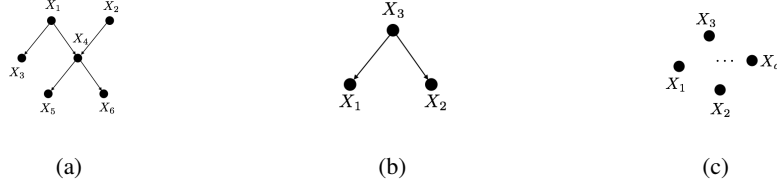

Figure 1: (a) An example of Bayesian Network $\mathcal{G}$ with $\overline{\mathbb{P}}_X$ as the induced distribution $\mathbb{P}_{X_1}\mathbb{P}_{X_2}\mathbb{P}_{X_3|X_1}$ $\mathbb{P}_{X_4|X_1,X_2}\mathbb{P}_{X_5|X_4}\mathbb{P}_{X_6|X_4}$. (b) A Bayesian Network $\mathcal{G}$ inducing a Markov chain $\mathbb{P}_{X_3}\mathbb{P}_{X_1|X_3}\mathbb{P}_{X_2|X_3}$. (c) A Bayesian Network $\mathcal{G}$ with $\overline{\mathbb{P}}_X$ as the induced distribution $\mathbb{P}_{X_1}\mathbb{P}_{X_2}\cdots\mathbb{P}_{X_d}$.

edge, these are the first family of estimators that are well defined for general probability spaces (breaking the $\Sigma H$ paradigm).

3. **Consistency proofs** (Section 4): We prove that the proposed estimators converge to the true value of the corresponding graph divergence measures as the number of observed samples increases in a general setting which includes several cases: (1) the data has some discrete and some continuous valued components (2) some (or all) of the components themselves are discrete-continuous mixtures (3) the data is real-valued but does not have a joint density on the entire space but is supported on a low-dimensional manifold.

4. **Numerical results** (Section 5): Extensive numerical results demonstrate that (1) existing algorithms have severe failure modes in general probability spaces (strongly negative values, for example), and (2) our proposed estimator achieves consistency as well as significantly better finite-sample performance.

## 2 Graph Divergence Measure

In this section, we define the family of graph divergence measures. To begin with, we first define some notational preliminaries. We denote any random variable by an uppercase letter such as $X$. The sample space of the variable $X$ is denoted by $\mathcal{X}$ and any value in $\mathcal{X}$ is denoted by the lowercase letter $x$. For any subset $A \subseteq \mathcal{X}$, the probability of $A$ for a given distribution function $\mathbb{P}_X(.)$ over $X$ is denoted by $\mathbb{P}_X(A)$. We note that the random variable $X$ can be a $d$-dimensional *vector* of random variables, i.e. $X \equiv (X_1, \ldots, X_d)$. The $N$ observed samples drawn from the distribution $\mathbb{P}_X$ are denoted by $x^{(1)}, x^{(2)}, \ldots, x^{(N)}$, i.e. $x^{(i)}$ is the $i^{th}$ observed sample.

Sometimes we might be interested in a subset of components of a random variable, $S \subseteq \{X_1, \ldots, X_d\}$ instead of the entire vector $X$. Accordingly, the sample space of the variable $S$ is denoted by $\mathcal{S}$. For instance, $X = (X_1, X_2, X_3, X_4)$ and $S = (X_1, X_2)$. Throughout the entire paper, unless otherwise stated, there is a one-to-one correspondence between the notations of $X$ and any $S$. For example for any value $x \in \mathcal{X}$, the corresponding value in $\mathcal{S}$ is simply denoted by $s$. Further, $s^{(i)} \in \mathcal{S}$ represents the lower-dimensional sample corresponding to the $i$th observed sample $x^{(i)} \in \mathcal{X}$. Furthermore, any marginal distribution defined over $S$ with respect to $\mathbb{P}_X$ is denoted by $\mathbb{P}_S$.

Consider a directed acyclic graph (DAG) $\mathcal{G}$ defined over $d$ nodes (corresponding to the $d$ components of the random variable $X$). A probability measure $Q$ over $X$ is said to be compatible with the graph $\mathcal{G}$ if it is a Bayesian network on $\mathcal{G}$. Given a graph $\mathcal{G}$ and a distribution $\mathbb{P}_X$, there is a natural measure $\overline{\mathbb{P}}_X(.)$ which is compatible with the graph and is defined as follows:

$$\overline{\mathbb{P}}_X = \prod_{l=1}^{d} \mathbb{P}_{X_l|\mathrm{pa}(X_l)} \tag{1}$$

where $\mathrm{pa}(X_l) \subset X$ is the set of the *parent* nodes of the random variable $X_l$, with the sample space denoted by $\mathcal{X}_{\mathrm{pa}(l)}$, and the sample values $x_{\mathrm{pa}(l)}$ corresponding to $x$. The distribution $\mathbb{P}_{X_l|\mathrm{pa}(X_l)}$ is the *conditional* distribution of $X_l$ given $\mathrm{pa}(X_l)$. Throughout the paper, whenever mentioning the variable $X_l$ with its own parents $\mathrm{pa}(X_l)$ we indicate it by $\mathrm{pa+}(X_l)$, i.e. $\mathrm{pa+}(X_l) \equiv \left(X_l, \mathrm{pa}(X_l)\right)$. An example is shown in Fig. 1a.

We note the fact that $\mathbb{P}_{S|X\setminus S}$ is well defined for any subset of variables $S \subset X$. Therefore if we let $S = X \setminus \mathrm{pa}(X_l)$, then $\mathbb{P}_{X\setminus\mathrm{pa}(X_l)|\mathrm{pa}(X_l)}$ is well defined for any $l \in \{1,\ldots,d\}$. By marginalizing over $X \setminus \mathrm{pa}+(X_l)$ we see that $\mathbb{P}_{X_l|\mathrm{pa}(X_l)}$ and thus the distribution $\overline{\mathbb{P}}_X$ is well defined.

The graph divergence measure is now defined as a function of the probability measure $\mathbb{P}_X$ and the graph $\mathcal{G}$. In this work we will focus only on the KL Divergence as being the distance metric, hence unless otherwise stated $D(\cdot \| \cdot) = D_{KL}(\cdot \| \cdot)$. Let's first consider the case where $\mathbb{P}_X$ is absolutely continuous with respect to $\overline{\mathbb{P}}_X$ and hence the *Radon-Nikodym* derivative $d\mathbb{P}_X/d\overline{\mathbb{P}}_X$ exists. Therefore for a given set of random variables $X$ and a Bayesian Network $\mathcal{G}$, we define **Graph Divergence Measure (**$\mathbb{GDM}$**)** as :

$$\mathbb{GDM}(X,\mathcal{G}) = D(\mathbb{P}_X \| \overline{\mathbb{P}}_X) = \int_{\mathcal{X}} \log \frac{d\mathbb{P}_X}{d\overline{\mathbb{P}}_X} d\mathbb{P}_X \tag{2}$$

Here we implicitly assume that $\log\left(d\mathbb{P}_X/d\overline{\mathbb{P}}_X\right)$ is measurable and integrable with respect to the measure $\mathbb{P}_X$. The $\mathbb{GDM}$ is set to infinity wherever Radon-Nikodym derivative does not exist. It is clear that $\mathbb{GDM}(X,\mathcal{G}) = 0$ if and only if the data distribution is compatible with the graphical model, thus the $\mathbb{GDM}$ can be thought of as a *measure of incompatibility with the given graphical model structure*.

We now have relevant variational characterization as below on our graph divergence measure, which can be harnessed to compute upper and lower bounds (More details in **supplementary material**):

**Proposition 2.1.** *For a random variable $X$, a DAG $\mathcal{G}$, let $\Pi(\mathcal{G})$ be the set of measures $\mathbb{Q}_X$ defined on the Bayesian Network $\mathcal{G}$, then $\mathbb{GDM}(X,\mathcal{G}) = \inf_{\mathbb{Q}_X \in \Pi(\mathcal{G})} D(\mathbb{P}_X \| \mathbb{Q}_X)$.*

*Furthermore, let $\mathcal{C}$ denote the set of functions $h : \mathcal{X} \to \mathbb{R}$ such that $\mathbb{E}_{\mathbb{Q}_X}[\exp(h(X))] < \infty$. If $\mathbb{GDM}(X,\mathcal{G}) < \infty$, then for every $h \in \mathcal{C}$, $\mathbb{E}_{\mathbb{P}_X}[h(X)]$ exists and:*

$$\mathbb{GDM}(X,\mathcal{G}) = \sup_{h\in\mathcal{C}} \mathbb{E}_{\mathbb{P}_X}[h(X)] - \log \mathbb{E}_{\mathbb{Q}_X}[\exp(h(X))]. \tag{3}$$

## 2.1 Special cases

For specific choices of $X$ and Bayesian Network, $\mathcal{G}$, the Equation 2 is reduced to the well-known information measures. Some examples of these measures are as follows:

**Mutual Information (MI):** $X = (X_1, X_2)$ and $\mathcal{G}$ has no directed edge between $X_1$ and $X_2$. Thus $\overline{\mathbb{P}}_X = \mathbb{P}_{X_1}.\mathbb{P}_{X_2}$, and we get, $\mathbb{GDM}(X,\mathcal{G}) = I(X_1; X_2) = D(\mathbb{P}_{X_1 X_2} \| \mathbb{P}_{X_1}\mathbb{P}_{X_2})$.

**Conditional Mutual Information (CMI):** We recover the conditional mutual information of $X_1$ and $X_2$ given $X_3$ by constraining $\mathcal{G}$ to be the one in Fig. 1b, since $\overline{\mathbb{P}}_X = \mathbb{P}_{X_3}.\mathbb{P}_{X_2|X_3}.\mathbb{P}_{X_1|X_3}$, i.e., $\mathbb{GDM}(X,\mathcal{G}) = I(X_1; X_2|X_3) = D(\mathbb{P}_{X_1 X_2 X_3} \| \mathbb{P}_{X_1|X_3}\mathbb{P}_{X_2|X_3}\mathbb{P}_{X_3})$.

**Total Correlation (TC):** When $X = (X_1, \cdots, X_d)$, and $\mathcal{G}$ is the graph with no edges (as in Fig. 1c, we recover the total correlation of the random variables $X_1, \ldots, X_d$ since $\overline{\mathbb{P}}_X = \mathbb{P}_{X_1} \ldots \mathbb{P}_{X_d}$, i.e., $\mathbb{GDM}(X,\mathcal{G}_{dc}) = TC(X_1, \ldots, X_d) = D(\mathbb{P}_{X_1\ldots X_d} \| \mathbb{P}_{X_1} \ldots \mathbb{P}_{X_d})$

**Multivariate Mutual Information (MMI) :** While the MMI defined by [9] is not positive in general,there is a different definition by [10] which is both non-negative and has an operational interpretation. Since MMI can be defined as the optimal total correlation after clustering, we can utilize our definition to define MMI (cf. **supplementary material**).

**Directed Information :** Suppose there are two stationary random processes $X$ and $Y$, the directed information rate from $X$ to $Y$ as first introduced by Massey [31] is defined as:

$$I(X \to Y) = \frac{1}{T} \sum_{t=1}^{T} I\left(X^t; Y_t | Y^{t-1}\right)$$

It can be seen that the directed information can be written as:

$$I(X \to Y) = \mathbb{GDM}\left((X^T, Y^T), \mathcal{G}_I\right) - \mathbb{GDM}\left((X^T, Y^T), \mathcal{G}_C\right)$$

where the graphical model $\mathcal{G}_I$ correponds to the *independent* distribution between $X^T$ and $Y^T$, and $\mathcal{G}_C$ corresponds to the *causal* distribution from $X$ to $Y$ (more details provided in **supplementary material**).

# 3 Estimators

## 3.1 Prior Art

Estimators for entropy date back to Shannon, who guesstimated the entropy rate of English [32]. Discrete entropy estimation is a well-studied topic and minimax rate of this problem is well-understood as a function of the alphabet size [33–35]. The estimation of differential entropy is considerably harder and also studied extensively in literature [23,25,26,36–39] and can be broadly divided into two groups; based on either Kernel density estimates [40,41] or based on k-nearest-neighbor estimation [27,42,43]. In a remarkable work, Kozachenko and Leonenko suggested a nearest neighbor method for entropy estimation [22] which was then generalized to a $k$th nearest neighbor approach [44]. In this method, the distance to the $k$th nearest neighbor (KNN) is measured for each data-point, and based on this the probability density around each data point is estimated and substituted into the entropy expression. When $k$ is fixed, each density estimate is noisy and the estimator accrues a bias and a remarkable result is that the bias is distribution-independent and can be subtracted out [45].

While the entropy estimation problem is well-studied, mutual information and its generalizations are typically estimated using a sum of signed entropy ($H$) terms, which are estimated first; we term such estimators as $\Sigma H$ estimators. In the discrete alphabet case, this principle has been shown to be worst-case optimal [28]. In the case of distributions with a joint density, an estimator that breaks the $\Sigma H$ principle is the KSG estimator [46], which builds on the KNN estimation paradigm but couples the estimates in order to reduce the bias. This estimator is widely used and has excellent practical performance. The original paper did not have any consistency guarantees and its convergence rates were recently established [47]. There have been some extensions to the KSG estimator for other information measures such as conditional mutual information [48,49], directed information [50] but none of them show theoretical guarantees on consistency of the estimators, furthermore they fail completely in mixture distributions.

When the data distribution is neither discrete nor admits a joint density, the $\Sigma H$ approach is no longer feasible as the individual entropy terms are not well defined. This is the regime of interest in our paper. Recently, Gao et al [51] proposed a mutual-information estimator based on KNN principle, which can handle such continuous-discrete mixture cases, and the consistency was demonstrated. However it is not clear how it generalizes to even Conditional Mutual Information (CMI) estimation, let alone other generalizations of mutual information. In this paper, we build on that estimator in order to design an estimator for general graph divergence measures and establish its consistency for generic probability spaces.

## 3.2 Proposed Estimator

The proposed estimator is given in Algorithm 1 where $\psi(\cdot)$ is the digamma function and $\mathbf{1}_{\{.\}}$ is the indicator function. The process is schematically shown in Fig. 3 (cf. **supplementary material**). We used the $\ell_\infty$-norm everywhere in our algorithm and proofs.

The estimator intuitively estimates the $\mathbb{GDM}$ by the *resubstitution estimate* $\frac{1}{N}\sum_{i=1}^{N}\log\hat{f}(x^{(i)})$ in which $\hat{f}(x^{(i)})$ is the estimation of Radon-Nikodym derivative at each sample $x^{(i)}$. If $x^{(i)}$ lies in a region where there is a density, the RN derivative is equal to $g_X(x^{(i)})/\bar{g}_X(x^{(i)})$ in which $g_X(.)$ and $\bar{g}_X(.)$ are density functions corresponding to $\mathbb{P}_X$ and $\overline{\mathbb{P}}_X$ respectively. On the other hand, if $x^{(i)}$ lies on a point where there is a discrete mass, the RN derivative will be equal to $h_X(x^{(i)})/\bar{h}_X(x^{(i)})$ in which $h_X(.)$ and $\bar{h}_X(.)$ are mass functions corresponding to $\mathbb{P}_X$ and $\overline{\mathbb{P}}_X$ respectively.

The density function $\bar{g}_X(x^{(i)})$ can be written as $\prod_{l=1}^{d}\left(g_{\mathrm{pa+}(X_l)}(x_{\mathrm{pa+}(l)}{}^{(i)})/g_{\mathrm{pa}(X_l)}(x_{\mathrm{pa}(l)}{}^{(i)})\right)$ for continuous components. Equivalently, the mass function $\bar{h}_X(x^{(i)})$ can be written as $\prod_{l=1}^{d}\left(h_{\mathrm{pa+}(X_l)}(x_{\mathrm{pa+}(l)}{}^{(i)})/h_{\mathrm{pa}(X_l)}(x_{\mathrm{pa}(l)}{}^{(i)})\right)$. Thus we need to estimate the density functions $g(.)$ and the mass functions $h(.)$ according to the type of $x^{(i)}$. The existing $k$th nearest neighbor algorithms will suffer while estimating the mass functions $h(.)$, since $\rho_{n_S,i}$ (the distance to the $n_S$-th nearest neighbor in subspace $\mathcal{S}$) for such points will be equal to zero for large $N$. Our algorithm, however, is designed in a way that it's capable of approximating both $g(.)$ functions as $\approx\frac{n_S}{N}\frac{1}{(\rho_{n_S,i})^{d_S}}$ and $h(.)$ functions as $\approx\frac{n_S}{N}$ dynamically for any subset $S\subseteq X$. This is achieved by setting $\rho_{n_S,i}$ terms such that all of them cancel out, yielding the estimator as in Eq. (4).

.

**Input: Parameter:** $k \in \mathbf{Z}^+$, **Samples:** $x^{(1)}, x^{(2)}, \ldots, x^{(N)}$, **Bayesian Network:** $\mathcal{G}$ on **Variables:**
$\quad \mathcal{X} = (X_1, X_2, \cdots, X_d)$
**Output:** $\widehat{\mathbb{GDM}}^{(N)}(X, \mathcal{G})$
1: **for** $i = 1$ to $N$ **do**
2: $\quad$ **Query:**
3: $\quad$ $\rho_{k,i} = \ell_\infty$-distance to the $k$th nearest neighbor of $x^{(i)}$ in the space $\mathcal{X}$
4: $\quad$ **Inquire:**
5: $\quad$ $\tilde{k}_i$ = # points within the $\rho_{k,i}$-neighborhood of $x^{(i)}$ in the space $\mathcal{X}$
6: $\quad$ $n_{\mathrm{pa}(X_l)}^{(i)}$ = # points within the $\rho_{k,i}$-neighborhood of $x^{(i)}$ in the space $\mathcal{X}_{\mathrm{pa}(l)}$
7: $\quad$ $n_{\mathrm{pa+}(X_l)}^{(i)}$ = # points within the $\rho_{k,i}$-neighborhood of $x^{(i)}$ in the space $\mathcal{X}_{\mathrm{pa+}(l)}$
8: $\quad$ **Compute:**
9: $\quad$ $\zeta_i = \psi(\tilde{k}_i) + \sum_{l=1}^{d} \left( \mathbf{1}_{\{\mathrm{pa}(X_l) \neq \emptyset\}} \log \left( n_{\mathrm{pa}(X_l)}^{(i)} + 1 \right) - \log \left( n_{\mathrm{pa+}(X_l)}^{(i)} + 1 \right) \right)$
10: **end for**
11: Final Estimator:

$$\widehat{\mathbb{GDM}}^{(N)}(X, \mathcal{G}) = \frac{1}{N} \sum_{i=1}^{N} \zeta_i + \left( \sum_{l=1}^{d} \mathbf{1}_{\{\mathrm{pa}(X_l) = \emptyset\}} - 1 \right) \log N \qquad (4)$$

Algorithm 1: Estimating **Graph Divergence Measure** $\mathbb{GDM}(X, \mathcal{G})$

## 4 Proof of Consistency

The proof of consistency for our estimator consists of two steps: First we prove that the expected value of the estimator in Eq. (4) converges to the true value as $N \to \infty$, and second we prove that the variance of the estimator converges to zero as $N \to \infty$. Let's begin with the definition of $P_X(x, r)$:

$$P_X(x, r) = \mathbb{P}_X \big\{ a \in \mathcal{X} : \|a - x\|_\infty \leq r \big\} = \mathbb{P}_X \Big\{ B_r(x) \Big\} \qquad (5)$$

Thus $P_X(x, r)$ is the probability of a hypercube with the edge length of $2r$ centered at the point $x$. We then state the following assmuptions:

**Assumption 1.** *We make the following assumptions to prove the consistency of our method:*

1. *$k$ is set such that $\lim_{N \to \infty} k = \infty$ and $\lim_{N \to \infty} \frac{k \log N}{N} = 0$.*

2. *The set of discrete points $\{x : P_X(x, 0) > 0\}$ is finite.*

3. *$\int_{\mathcal{X}} \big| \log f(x) \big| d\mathbb{P}_X < +\infty$, where $f \equiv d\mathbb{P}_X / d\overline{\mathbb{P}}_X$ is Radon-Nikodym derivative.*

The Assumption 1.1 with 1.2 controls the boundary effect between the continuous and the discrete regions; with this assumption we make sure that all the $k$ nearest neighbors of each point belong to the same region almost surely (i.e. all of them are either continuous or discrete). Assumption 1.3 is the log-integrability of the Radon-Nikodym derivative. These assumptions are satisfied under mild technical conditions whenever the distribution $\mathbb{P}_X$ over the set $\mathcal{X}$ is (1) finitely discrete; (2) continuous; (3) finitely discrete over some dimensions and continuous over some others; (4) a mixture of the previous cases; (5) has a joint density supported over a lower dimensional manifold. These cases represent almost all the real world data.

As an example of a case not conforming to the aforementioned cases, we can consider singular distributions, among which the *Cantor distribution* is a significant example whose cumulative distribution function is the Cantor function. This distribution has neither a probability density function nor a probability mass function, although its cumulative distribution function is a continuous function. It is thus neither a discrete nor an absolutely continuous probability distribution, nor is it a mixture of these.

The Theorem 1 formally states the mean-convergence of the estimator while Theorem 2 formally states that convergence of the variance to zero.

**Theorem 1.** *Under the Assumptions 1, we have* $\lim_{N \to \infty} \mathbb{E}\left[\widehat{\mathbb{GDM}}^{(N)}(X, \mathcal{G})\right] = \mathbb{GDM}(X, \mathcal{G})$.

**Theorem 2.** *In addition to the Assumptions 1, assume that we have* $(k_N \log N)^2 / N \to 0$ *as $N$ goes to infinity. Then we have* $\lim_{N \to \infty} Var\left[\widehat{\mathbb{GDM}}^{(N)}(X, \mathcal{G})\right] = 0$.

The Theorems 1 and 2 combined yield the consistency of the estimator 4.

The proof of the Theorem 1 starts with writing the Radon-Nikodym derivative explicitly. Then we need to upper-bound the term $\left|\mathbb{E}\left[\widehat{\mathbb{GDM}}^{(N)}(X, \mathcal{G})\right] - \mathbb{GDM}(X, \mathcal{G})\right|$. To achieve this goal, we segregate the domain of $\mathcal{X}$ into three parts as $\mathcal{X} = \Omega_1 \cup \Omega_2 \cup \Omega_3$ where $\Omega_1 = \{x : f(x) = 0\}$, $\Omega_2 = \{x : f(x) > 0, P_X(x, 0) > 0\}$ and $\Omega_3 = \{x : f(x) > 0, P_X(x, 0) = 0\}$. We will show that $\mathbb{P}_X(\Omega_1) = 0$. The sets $\Omega_2$ and $\Omega_3$ correspond to the discrete and continuous regions respectively. Then for each of the two regions, we introduce an upperbound which goes to zero as $N$ grows boundlessly. Thus equivalently we show the mean of the estimate $\zeta_1$ is close to $\log f(x)$ for any $x$.

The proof of the Theorem 2 is based on the Efron-Stein inequality, which upperbounds any estimator for any quantity from the observed samples $x^{(1)}, \ldots, x^{(N)}$. For any sample $x^{(i)}$, we then upperbound the term $\left|\zeta_i(X) - \zeta_i(X_{\setminus j})\right|$ by segregating the samples into various cases, and examining each case separately. $\zeta_i(X)$ is the estimate using all the samples $x^{(1)}, \ldots, x^{(N)}$ and $\zeta_i(X_{\setminus j})$ is the estimate when the $j$th sample is removed. Summing up over all the $i$'s, we obtain an upper-bound which will converge to 0 as $N$ goes to infinity.

## 5 Empirical Results

In this section, we evaluate the performance of our proposed estimator in comparison with other estimators via numerical experiments. The estimators evaluated here are our estimator referred to as *GDM*, the plain KSG-based estimators for continuous distributions to which we refer by *KSG*, the *binning* estimators and the noise-induced $\Sigma H$ estimators. A more detailed discussion can be found in Section G.

**Experiment 1: Markov chain model with continuous-discrete mixture.** For the first experiment, we simulated an $X$-$Z$-$Y$ Markov chain model in which the random variable $X$ is a uniform random variable $\mathcal{U}(0, 1)$ clipped at a threshold $0 < \alpha_1 < 1$ from above. Then $Z = \min(X, \alpha_2)$ and $Y = \min(Z, \alpha_3)$ in which $0 < \alpha_3 < \alpha_2 < \alpha_1$. We simulated this system for various numbers of samples, setting $\alpha_1 = 0.9$, $\alpha_2 = 0.8$ and $\alpha_3 = 0.7$. For each set of samples we estimated $I(X; Y|Z)$ via different methods. The theory value for $I(X; Y|Z)$ is 0. The results are shown in Figure 2a. We can see that in this regime, only the GDM estimator can correctly converge. The KSG estimator and the $\Sigma H$ estimator show high negative biases and the binning estimator shows a positive bias.

**Experiment 2: Mixture of AWGN and BSC channels with variable error probability.** For the second scheme of our experiments, we considered an Additive White Gaussian Noise (AWGN) Channel in parallel with a Binary Symmetric Channel (BSC) where only one of the two can be activated at a time. The random variable $Z = \min(\alpha, \tilde{Z})$ where $\tilde{Z} \sim U(0, 1)$ controls which channel is activated; i.e. if $Z$ is lower than the threshold $\beta$, activate the AWGN channel, otherwise initiate the BSC channel where $Z$ also determines the error probability at each time point. We set $\alpha = 0.3$, $\beta = 0.2$, BSC channel input as $X \sim \text{Bern}(0.5)$, and AWGN input and noise deviation as $\sigma_X = 1$ and $\sigma_N = 0.1$ respectively, and obtained the estimates of $I(X; Y|Z, Z^2, Z^3)$ for various estimators. While the theory value is equal to $I(X; Y|Z) = 0.53241$, yet it's conditioned over a low-dimensional manifold in a high-dimensional space. The results are shown in Figure 2b. Similar to the previous experiment, the GDM estimator can correctly converge to the true value. The $\Sigma H$ and binning estimators show a negative bias, and the KSG estimator gets totally lost.

**Experiment 3: Total Correlation for independent mixtures.** In this experiment, we estimate the total correlation of three independent variables $X$, $Y$ and $Z$. The samples for the variable $X$ are generated in the following fashion: First toss a fair coin, if heads appears we fix $X$ at $\alpha_X$, otherwise we draw $X$ from a uniform distribution between 0 and 1. samples from $Y$ and $Z$ are also generated in the same way independently with parameters $\alpha_Y$ and $\alpha_Z$ respectively. For this setup, $TC(X, Y, Z) = 0$. We set $\alpha_X = 1$, $\alpha_Y = 1/2$ and $\alpha_Z = 1/4$, and generated various datasets with different lengths. Then estimated total correlation values are shown in the Figure 2c.

**Experiment 4: Total Correlation for independent uniforms with correlated zero-inflation.** Here we first consider four auxiliary uniform variables $\tilde{X}_1$, $\tilde{X}_2$, $\tilde{X}_3$ and $\tilde{X}_4$ which are taken from $\mathcal{U}(0.5, 1.5)$. Then each sample is erased with a Bernoulli probability; i.e. $X_1 = \alpha_1 \tilde{X}_1$, $X_2 = \alpha_1 \tilde{X}_2$ and $X_3 = \alpha_2 \tilde{X}_3$, $X_4 = \alpha_2 \tilde{X}_4$ in which $\alpha_1 \sim \text{Bern}(p_1)$ and $\alpha_2 \sim \text{Bern}(p_2)$. As we see, after zero-inflation $X_1$ and $X_2$ become correlated, so do $X_3$ and $X_4$ while still $(X_1, X_2) \perp\!\!\!\perp (X_3, X_4)$. In the experiment, we set $p_1 = p_2 = 0.6$. The results of running different algorithms over the data can be seen in Figure 2d. For the total correlation experiments 3 and 4, similar to that of conditional mutual information in experiments 1 and 2, only the GDM estimator can best estimate the true value. The estimator $\Sigma H$ was removed from the figures due to its high inaccuracy.

**Experiment 5: Gene Regulatory Networks.** In this experiment we use different estimators to do Gene Regulatory Network inference based on the conditional Restricted Directed Information (cRDI) [20]. We do our test on the simulated neuron cells' development process, based on a model from [52]. In this model, the time series vector $X$ consists of 13 random variables each of which corresponding to a single gene in the development process. We simulated the development process for various lengths of time-series in which the noise $N \sim \mathcal{N}(0, .03)$ is added for all the genes, and every single sample is then subject to erasure (i.e. be replaced by 0s) with a probability of $0.5$. Then we applied the cRDI method utilizing various CMI estimators and then calculated the Area-Under-ROC curve (AUROC). The results are shown in Figure 2e. It's seen that the cRDI method implemented with the GDM estimator outperform the other estimators by at least $\%10$ in terms of AUROC. In the tests, cRDI for each $(X_i, X_j)$ is conditioned over the node $k \neq i$ with the highest RDI value to $j$. We notice that the causal signals are highly destroyed due to the zero-inflation, so we won't expect high performance of the causal inference over the data. We did not include the $\Sigma H$ estimator results due to its very low performance.

**Experiment 6: Feature Selection by Conditional Mutual Information Maximization.** Feature selection is an important pre-processing step in many learning tasks. The application of information theoretic measures in feature selection is well studied in the literature [7]. Among the well-known methods is the conditional mutual information maximization (CMIM) first introduced by Flueret [4], a variation of which was later introduced called CMIM-2 [53]. Both methods use conditional mutual information as their core measure to select the features. Hence the performance of the estimators can significantly influence the performance of the methods. In our experiment, we generated a vector $X = (X_1, \ldots, X_{15})$ of 15 random variables in which all the random variables are taken from $\mathcal{N}(0, 1)$ and then each random variable $X_i$ is clipped from above at $\alpha_i$ which is initially taken randomly from $\mathcal{U}(0.25, 0.3)$ and then kept constant during the sample generation. Then $Y$ is generated as $Y = \cos\left(\sum_{i=1}^{5} X_i\right)$. Then we did the CMIM-2 algorithm with various CMI estimators to evaluate the performance of the estimators in extracting the relevant features $X_1, \ldots, X_5$. The AUROC values for each algorithm versus the number of samples generated are shown in Figure 2f. The feature selection methods implemented with the GDM estimator outperform the other estimators.

# 6  Discussion and Future Work

A general paradigm of graph divergence measures and novel estimators thereof, for general probability spaces are proposed, which estimate several generalizations of mutual information. In future, we would like to derive further efficient estimators for high dimensional data. In the current work, estimators are shown to be consistent with infinite scaling of parameter $k$. In future, we would like to understand the finite $k$ performance of the estimators as well as guarantees on sample complexity and rates of convergence. Another potential direction to follow is to study the variational characterization of the graph divergence measure to design estimators. Improving the computational efficiency of the estimator is another direction of future work. Recent literature including [54] provide a new methodology to estimate mutual information in a computationally efficient manner and leveraging these ideas for the generalized measures and general proability distributions can be a promising direction ahead.

# 7  Acknowledgement

This work was partially supported by NSF grants 1651236, 1703403 and NIH grant 5R01HG008164. The authors also would like to thank Yihan Jiang for presenting our work at the NeurIPS conference.

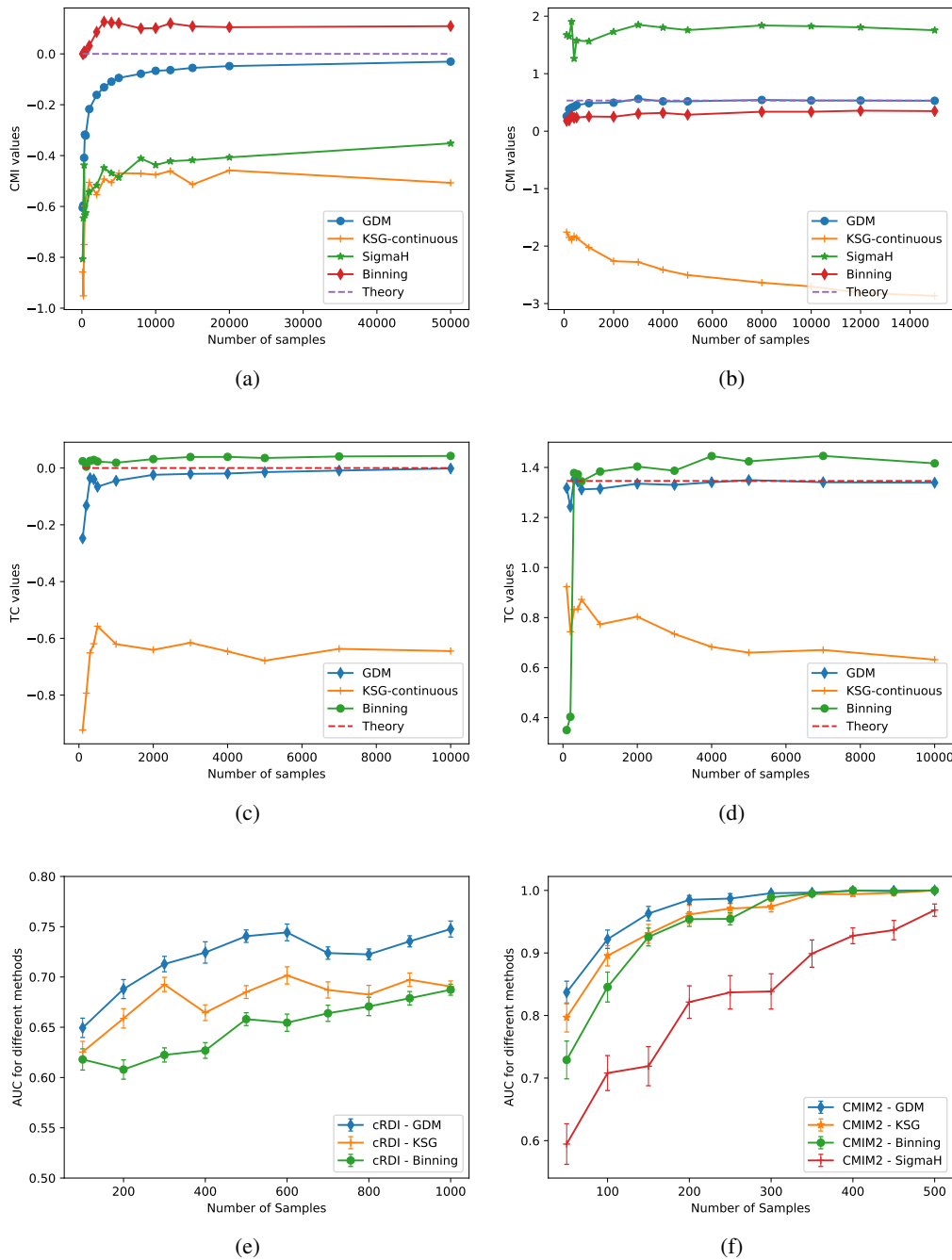

Figure 2: The results for the experiments versus the number of samples: 2a: The estimated CMI for the X-Z-Y Markov chain. 2b: CMI for the AWGN+BSC channels with low-dimensional $Z$ manifold. 2c: The estimated TC values for three independent variables. 2d: The estimated TC for zero-inflated variables. 2e: The AUROC values for gene regulatory network inference. The error bars show the standard deviation scaled down by $0.2$. 2f: The AUROC values for feature selection accuracy. The error bars show the standard deviations scaled down by $0.2$.

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
