[Supplementary Material]



Figure 3: The method for estimating information theoretic measures. **(a)** Step 1: **Query** for the distance to the $k$th nearest neighbor of each point $i$ in the space $\mathcal{X}$. **(b)** Step 2: **Inquire** for the number of points lying within the $\rho_{k,i}$-neighborhood of each point $i$ in the subspaces of $\mathcal{X}$ including itself.

## A  Pictorial Representation of Estimators

The Figure 3a represents the step 1 (distance query or **Query** step in the algorithm) and the Figure 3b represents the steps 2 and 3 (numbers inquiry, or the **Inquire** step in the algorithm). Note that in the graphics we used $\ell_2$-norm to give better intuition on the process, while in our proofs and simulations, we use $\ell_\infty$-norm.

## B  Multivariate Mutual Information

In [10], Multivariable Mutual Information (MMI) is defined as follows: let $\Pi(\mathcal{X})$ be the collection of all possible partitions of $\mathcal{P}$ which split $\mathcal{X}$ into at least two non-empty disjoint subsets. For any partition $\mathcal{P} \in \Pi(\mathcal{X})$, the product distribution $\Pi_{C \in \mathcal{P}} \mathbb{P}_{X_C}$ specifies an independence relation, i.e., $X_C$'s are treated as agglomerated random variables and are mutually independent. Given a particular partition, define an information measure $I_{\mathcal{P}}(X)$ as :

$$I_{\mathcal{P}}(X) = \frac{1}{\mathcal{P} - 1} D(\mathbb{P}_X \parallel \Pi_{C \in \mathcal{P}} \mathbb{P}_{X_C}) \tag{6}$$

Then, MMI is defined as:

$$\text{MMI(X)} = \min_{\mathcal{P} \in \Pi(\mathcal{X})} I_{\mathcal{P}}(X) \tag{7}$$

This can cast as a functional of our Graph Divergence Measure by choosing for every partition, $\mathcal{P} \in \Pi(\mathcal{X})$, a DAG $\mathcal{G}_{\mathcal{P}}$ with all $X_C$'s forming an aggregate node but disconnected from each other and thus inducing a measure $\overline{\mathbb{P}}_X^{\mathcal{P}}$. Thus,

$$I_{\mathcal{P}}(X) = \frac{1}{\mathcal{P} - 1} \mathbb{GDM}(\mathbb{P}_X \parallel \overline{\mathbb{P}}_X^{\mathcal{P}}) \tag{8}$$

which implies,

$$\text{MMI(X)} = \min_{\mathcal{P} \in \Pi(\mathcal{X})} \mathbb{GDM}(\mathbb{P}_X \parallel \overline{\mathbb{P}}_X^{\mathcal{P}}) \tag{9}$$

## C  Directed Information

In this section, we will derive the expression of the directed information from $X^T = (X_1, \ldots, X_T)$ to $Y^T = (Y_1, \ldots, Y_T)$ in terms of two graph divergence measures. For the simplicity of notations and logic we will only do it assuming $X$ and $Y$ are discrete. However, it's easily extendable to the

case of mixture distributions using the notion of Radon-Nikodym derivative. From the definition of the directed information we have:

$$I\left(X^T \to Y^T\right) \tag{10}$$

$$= \sum_{t=1}^{T} I\left(X^t; Y_t | Y_{t-1}\right) \tag{11}$$

$$= \sum_{t=1}^{T} \sum_{x^t, y^t} \left( \mathbb{P}_{X^t Y^t}(x^t, y^t) \log \frac{\mathbb{P}_{Y_t | X^t, Y^{t-1}}(y_t | x^t, y^{t-1})}{\mathbb{P}_{Y_t | Y^{t-1}}(y_t | y^{t-1})} \right) \tag{12}$$

$$= \sum_{x^T, y^T} \mathbb{P}_{X^T Y^T}(x^T, y^T) \sum_{t=1}^{T} \log \frac{\mathbb{P}_{Y_t | X^t, Y^{t-1}}(y_t | x^t, y^{t-1})}{\mathbb{P}_{Y_t | Y^{t-1}}(y_t | y^{t-1})} \tag{13}$$

$$= \sum_{x^T, y^T} \mathbb{P}_{X^T Y^T}(x^T, y^T) \log \frac{\prod_{t=1}^{T} \mathbb{P}_{Y_t | X^t, Y^{t-1}}(y_t | x^t, y^{t-1})}{\prod_{t=1}^{T} \mathbb{P}_{Y_t | Y^{t-1}}(y_t | y^{t-1})} \tag{14}$$

$$= \sum_{x^T, y^T} \mathbb{P}_{X^T Y^T}(x^T, y^T) \log \frac{\prod_{t=1}^{T} \mathbb{P}_{Y_t | X^t, Y^{t-1}}(y_t | x^t, y^{t-1})}{\mathbb{P}_{Y^T}(y^T)} \tag{15}$$

$$= \sum_{x^T, y^T} \mathbb{P}_{X^T Y^T}(x^T, y^T) \log \frac{\prod_{t=1}^{T} \mathbb{P}_{Y_t | X^t, Y^{t-1}}(y_t | x^t, y^{t-1}) \mathbb{P}_{X_t | X^{t-1}}(x_t | x^{t-1})}{\mathbb{P}_{X^T}(x^T) \mathbb{P}_{Y^T}(y^T)} \tag{16}$$

$$= \sum_{x^T, y^T} \mathbb{P}_{X^T Y^T}(x^T, y^T) \log \frac{\mathbb{P}_{X^T Y^T}(x^T, y^T)}{\mathbb{P}_{X^T}(x^T) \mathbb{P}_{Y^T}(y^T)}$$

$$\quad - \sum_{x^T, y^T} \mathbb{P}_{X^T Y^T}(x^T, y^T) \log \frac{\mathbb{P}_{X^T Y^T}(x^T, y^T)}{\prod_{t=1}^{T} \mathbb{P}_{Y_t | X^t, Y^{t-1}}(y_t | x^t, y^{t-1}) \mathbb{P}_{X_t | X^{t-1}}(x_t | x^{t-1})} \tag{17}$$

$$= D\left(\mathbb{P}_{X^T Y^T} \| \mathbb{P}^I_{X^T Y^T}\right) - D\left(\mathbb{P}_{X^T Y^T} \| \mathbb{P}^C_{X^T Y^T}\right) \tag{18}$$

$$= \mathbb{GDM}\left((X^T, Y^T), \mathcal{G}_I\right) - \mathbb{GDM}\left((X^T, Y^T), \mathcal{G}_C\right) \tag{19}$$

The distributions $\mathbb{P}^I_{X^T Y^T} = \mathbb{P}_{X^T} \mathbb{P}_{Y^T}$ and $\mathbb{P}^C_{X^T Y^T} = \prod_{t=1}^{T} \mathbb{P}_{Y_t | X^t, Y^{t-1}} \mathbb{P}_{X_t | X^{t-1}}$ represent the *independent* distribution between $X^T$ and $Y^T$, and the *causal* distribution from $X^T$ to $Y^T$ respectively.

## D  Variational Representation of Graph Divergence Measure

The first part follow from the following property of KL divergence:

$$D(\mathbb{P}_X \| \mathbb{Q}_X) = D(\mathbb{P}_X \| \overline{\mathbb{P}}_X) + \sum_{j}^{d} D(\mathbb{P}_{X_j | X_{Pa(j)}} \| \mathbb{Q}_{X_j | X_{Pa(j)}} | \mathbb{P}_{X_1^{j-1}}) \tag{20}$$

The second part follows from the standard Donsker-Varadhan characterizations of vanilla divergence as in [55].

## E  Investigating assumptions in Theorem 1

In this section, we will investigate the validity of the assumptions for various types of distributions. In particular, we will investigate the different types of distributions we had mentioned: (1) finitely discrete; (2) continuous; (3) finitely discrete over some dimensions while continuous over others; (4) a mixture of the previous cases; (5) has a joint density supported over a lower dimensional manifold. As we had mentioned, these cases represent almost all the real world data. We note that the estimator is well defined only when $\mathcal{X} = \mathbb{R}^{d_X}$; i.e. all the alphabets be in real space.

We will examine the Assumptions 1.2 and 1.3 for each case separately. The Assumption 1.1 is a parametric assumption related to the convergence of the algorithm and is not directly related to the distribution of the data. Jointly considered with the Assumption 1.2, it controls the boundary effects between the continuous and the discrete regions.

## E.1 Finitely discrete distribution

For a *finitely discrete* distribution the Assumption 1.2 holds by definition. The Assumption 1.3 trivially holds since the size of sample space $|\mathcal{X}|$ is finite.

## E.2 Continuous distribution

For a continuous distribution, the Assumption 1.2 holds since there is no discrete component.

The distribution is absolutely continuous with respect to the *Lebesgue* measure $\lambda$ meaning that $\mathbb{P}_X(A) = 0$ for any subset $A \subset \mathcal{X}$ implies $\lambda(A) = 0$. Thus we can conclude that there exists a density $g : \mathcal{X} \to \mathbb{R}^+$ such that $\mathbb{P}_X(A) = \int_A g \, d\lambda$. Naturally $\mathbb{P}_X(\mathcal{X}) = \int_{\mathcal{X}} g \, d\lambda = 1$.

Furthermore if the density function $g$ is bounded everywhere and the variable has a bounded support, the Assumption 1.3 is fulfilled.

## E.3 Finitely discrete over some dimensions while continuous over others

In this case, the variable set $X$ can be decomposed to two sets $X^D$ and $X^C$ representing the finitely discrete and continuous dimensions respectively. So for any realization of the discrete dimensions $X^D = x^D$ the probability mass function $h_{X^D}(x^D)$ exists, and a conditional density $g_{X^C|X^D}(x^C|x^D)$ is well defined. We can define an auxiliary function $P_{X^C|X^D}(x^C, r|x^D) \equiv \mathbb{P}_{X^C|X^D}\left\{a \in \mathcal{X}^C : \|a - x^C\|_\infty \leq r | X^D = x^D\right\}$. Thus we can write

$$P_X(x^C, x^D, r) = \mathbb{P}_X\left\{(a,b) \in \mathcal{X} : b = x^D, \|a - x^C\|_\infty \leq r\right\} = P_{X^C|X^D}(x^C, r|x^D)h_{X^D}(x^D) \tag{21}$$

We will show that if for any $x^D$ the continuous distribution over $X^C$ satisfies the Assumptions 1 as discussed in the Section E.2, then we can see that $\mathbb{P}_X$ will also satisfy the assumptions.

Let's define $f_{x^D}(x^C)$ for any fixed $x^D$ as:

$$f_{x^D}(x^C) = \lim_{r \to 0} P_{X^C|X^D}(x^C, r|x^D) \prod_{l=1}^{d} \frac{P_{\mathrm{pa}(X_l)^C|\mathrm{pa}(X_l)^D}\left(x_{\mathrm{pa}(l)}{}^C, r\middle|x_{\mathrm{pa}(l)}{}^D\right)}{P_{\mathrm{pa+}(X_l)^C|\mathrm{pa+}(X_l)^D}\left(x_{\mathrm{pa+}(l)}{}^C, r\middle|x_{\mathrm{pa+}(l)}{}^D\right)} \tag{22}$$

Assuming the limit exists everywhere. Thus the Radon-Nikodym Derivative $f$ can be written as:

$$
\begin{aligned}
f(x) &= \lim_{r \to 0} P_X(x^C, x^D, r) \prod_{l=1}^{d} \frac{P_{\mathrm{pa}(X_l)}\left(x_{\mathrm{pa}(l)}{}^C, x_{\mathrm{pa}(l)}{}^D, r\right)}{P_{\mathrm{pa+}(X_l)}\left(x_{\mathrm{pa+}(l)}{}^C, x_{\mathrm{pa+}(l)}{}^D, r\right)} & (23)\\
&= \lim_{r \to 0} P_{X^C|X^D}(x^C, r|x^D) \prod_{l=1}^{d} \frac{P_{\mathrm{pa}(X_l)^C|\mathrm{pa}(X_l)^D}\left(x_{\mathrm{pa}(l)}{}^C, r\middle|x_{\mathrm{pa}(l)}{}^D\right)}{P_{\mathrm{pa+}(X_l)^C|\mathrm{pa+}(X_l)^D}\left(x_{\mathrm{pa+}(l)}{}^C, r\middle|x_{\mathrm{pa+}(l)}{}^D\right)} \\
&\quad \times h_{X^D}(x^D) \prod_{l=1}^{d} \frac{h_{\mathrm{pa}(X_l)}\left(x_{\mathrm{pa}(l)}{}^D\right)}{h_{\mathrm{pa+}(X_l)}\left(x_{\mathrm{pa+}(l)}{}^D\right)} & (24)\\
&= f_{x^D}(x^C) \times h_{X^D}(x^D) \prod_{l=1}^{d} \frac{h_{\mathrm{pa}(X_l)}\left(x_{\mathrm{pa}(l)}{}^D\right)}{h_{\mathrm{pa+}(X_l)}\left(x_{\mathrm{pa+}(l)}{}^D\right)} & (25)
\end{aligned}
$$

Therefore for Assumption 1.3 we have:

$$
\begin{aligned}
\int_{\mathcal{X}} |\log f(x)| \, d\mathbb{P}_X &\leq \sum_{x^D} h_{X^D}(x^D) \int_{\mathcal{X}^C} \left|\log f_{x^D}(x^C)\right| g_{X^C|X^D}(x^C|x^D) dx^C \\
&\quad + \sum_{x^D} h_{X^D}(x^D) \left|\log h_{X^D}(x^D) \prod_{l=1}^{d} \frac{h_{\mathrm{pa}(X_l)}\left(x_{\mathrm{pa}(l)}{}^D\right)}{h_{\mathrm{pa+}(X_l)}\left(x_{\mathrm{pa+}(l)}{}^D\right)}\right| & (26)
\end{aligned}
$$

In which we used the fact that $\mathbb{E}\left[\log f(x)\right] = \mathbb{E}_{X^D}\left[\mathbb{E}\left[\log f(x^C, x^D)\big|X^D = x^D\right]\right]$. The first term above is upper-bounded since we assumed that continuous distribution over $X^C$ satisfies the Assumption 1.3 for any $x^D$. The second term is upper-bounded since it's a finitely discrete distribution as discussed in Section E.1. Thus $\int_{\mathcal{X}} |\log f(x)| \, d\mathbb{P}_X < \infty$ and the Assumption 1.3 holds.

### E.4 A mixture of the previous cases

In this case, we assume that the probability can be described as a linear combination of a continuous and a discrete distribution, i.e. without loss of generality we can assume that for any subset $A \subset \mathcal{X}$ the distribution can be described as $\mathbb{P}(A) = \alpha_C \mathbb{P}_X^C(A) + \alpha_D \mathbb{P}_X^D(A)$ in which $\alpha_C + \alpha_D = 1$. We note that the $\mathbb{P}_X^D$ does not represent a mass function here since the mass function is only defined over a discrete alphabet $\mathcal{X}^D = \{x_1, ..., x_m\} \subset \mathcal{X}$ while $\mathbb{P}_X^D$ needs to admit the continuous domain $\mathcal{X}$. Thus we relate $\mathbb{P}_X^D$ to the mass function $h_X^D(.)$ as $\mathbb{P}^D(A) = \sum_{x \in \mathcal{X}^D} \mathbf{1}_{\{x \in A\}} h_X^D(x)$. Furthermore we can see that for any $x \in \mathcal{X}^D$ and for $r$ small enough:

$$P_X^D(x, r) \equiv \mathbb{P}_X^D\left(B_r(x)\right) = h_X^D(x) \tag{27}$$

If $x \notin \mathcal{X}^D$ then for small enouth $r$ we have $P_X^D(x, r) = 0$.

The Assumption 1.2 holds since the discrete distribution $\mathbb{P}_X^D$ is finite, and hence the number of total discrete points will be finite.

For the Assumption 1.3 we have:

$$\int_{\mathcal{X}} |\log f(x)| \, d\mathbb{P}_X = \alpha_C \int_{\mathcal{X}} |\log f(x)| \, d\mathbb{P}_X^C + \alpha_D \sum_{x \in \mathcal{X}^D} h_X^D(x) \, |\log f(x)| \tag{28}$$

and if the continuous distribution complies with the assumptions mentioned in Section E.2 then the term above is upper-bounded and the Assumption 1.3 holds.

### E.5 A distribution with a joint density supported over a lower dimensional manifold

This case simply means that the probability distribution $\mathbb{P}_X$ in $\mathcal{X}$ can be mapped to a probability distribution $\mathbb{P}_Y$ in a lower-dimensional space $\mathcal{Y}$ where $d_Y < d_X$, via a one-to-one continuous function $h : \mathcal{X} \to \mathcal{Y}$. If the lower-dimnesional distribution $\mathbb{P}_Y$ is continuous complying with the properties discussed in the Section E.2, then it will preserve all the properties through the inverse mapping $h^{-1} : \mathcal{Y} \to \mathcal{X}$ and $\mathbb{P}_X$ hence $\mathbb{P}_X$ will satisfy the Assumptions 1. If the distribution has finite discrete components either as a discrete or as a mixture distribution, the finiteness of the components will be preserved as well and hence $\mathbb{P}_X$ will satisfy the Assumptions 1.

Therefore this category of distributions will satisfy the Assumptions 1 if the distribution $\mathbb{P}_Y$ satisfies them.

## F Consistency Proofs

### F.1 Proof of Theorem 1

First let's generalize the definition of $P_X(x, r)$ in Equation 5 to any subset $S \subseteq X$, i.e. for any point $s \in S$ we define:

$$P_S(s, r) = \mathbb{P}_S\{a \in \mathcal{S} : \|a - s\|_\infty \leq r\} = \mathbb{P}_S\{B_r(s)\} \tag{29}$$

Thus $P_S(s, r)$ is the probability of an $\ell_\infty$ ball of radius $r$ centered at $s$, or equivalently, a hypercube with the edge length of $2r$ centered at the point $s$.

To prove the asymptotic unbiasedness of the estimator, we will first write the Radon-Nikodym derivative in an explicit form via the following lemma.

**Lemma 3.** *For almost every $x \in \mathcal{X}$:*

$$\frac{d\mathbb{P}_X}{d\overline{\mathbb{P}}_X}(x) = f(x) = \lim_{r \to 0} P_X(x, r) \prod_{l=1}^{d} \frac{P_{pa(X_l)}\left(x_{pa(l)}, r\right)}{P_{pa+(X_l)}\left(x_{pa+(l)}, r\right)} \tag{30}$$

*Proof.* Please see the section F.2. □

Now notice that $\widehat{\mathbb{GDM}}^{(N)}(X, \mathcal{G}) = \frac{1}{N}\sum_{i=1}^{N}\zeta_i$ in which all the $\zeta_i$'s are identically distributed. Thus we have $\mathbb{E}[\widehat{\mathbb{GDM}}^{(N)}(X, \mathcal{G})] = \mathbb{E}[\zeta_1]$. Therefore, the bias can be written as:

$$\left| \mathbb{E}\left[ \widehat{\mathbb{GDM}}^{(N)}(X, \mathcal{G}) \right] - \mathbb{GDM}(X, \mathcal{G}) \right| = \left| \mathbb{E}_X \left[ \mathbb{E}[\zeta_1 | X] \right] - \int_{\mathcal{X}} \log f(X) d\mathbb{P}_X \right| \tag{31}$$

$$\leq \int_{\mathcal{X}} \left| \mathbb{E}[\zeta_1 | X] - \log f(X) \right| d\mathbb{P}_X \tag{32}$$

Now we will give upper bounds for $|\mathbb{E}[\zeta_1|X] - \log f(X)|$ for every $x \in \mathcal{X}$. Similar to the technique used by [51], we divide the domain of $\mathcal{X}$ into three parts as $\mathcal{X} = \Omega_1 \bigcup \Omega_2 \bigcup \Omega_3$ where:

- $\Omega_1 = \{x : f(x) = 0\}$
- $\Omega_2 = \{x : f(x) > 0, P_X(x, 0) > 0\}$
- $\Omega_3 = \{x : f(x) > 0, P_X(x, 0) = 0\}$

For each of the domains, we show that $\lim_{N\to\infty} \int_{\Omega_i} |\mathbb{E}[\zeta_1|X = x] - \log f(x)| d\mathbb{P}_X = 0$.

**For $x \in \Omega_1$:**

The probability of $\Omega_1$ is zero with respect to $\mathbb{P}_X$, since:

$$\mathbb{P}_X(\Omega_1) = \int_{\Omega_1} d\mathbb{P}_X = \int_{\Omega_1} f(x)d\overline{\mathbb{P}}_X = \int_{\Omega_1} 0 d\overline{\mathbb{P}}_X = 0 \tag{33}$$

In which the second equality is due to Lemma 3. Thus $\int_{\Omega_1} |\mathbb{E}[\zeta_1|X = x] - \log f(x)| d\mathbb{P}_X = 0$.

**For $x \in \Omega_2$:**

In this case $f(x)$ is obviously the same as $P_X(x, 0) \prod_{l=1}^{d} \frac{P_{\mathrm{pa}(X_l)}(x_{\mathrm{pa}(l)}, 0)}{P_{\mathrm{pa+}(X_l)}(x_{\mathrm{pa+}(l)}, 0)}$. We will first show that the probability of the $k$-nearest neighbor distance $\rho_{k,1}$ being non-zero is small, which means we will use the number of samples being equal to $x$ as $\tilde{k}_i$, and we will show that the mean of the estimate $\zeta_1$ is close to $\log f(x)$.

We notice that for $x$, the probability of $\rho_{k,1} > 0$ is equal to the probability that $x$ is observed at most $k - 1$ times. So it can be upper bounded as:

$$\mathbb{P}\left( \rho_{k,1} > 0 \Big| X = x \right) \tag{34}$$

$$= \sum_{m=0}^{k-1} \binom{N-1}{m} P_X(x, 0)^m \left( 1 - P_X(x, 0) \right)^{N-1-m} \tag{35}$$

$$\leq \sum_{m=0}^{k-1} N^m \left( 1 - P_X(x, 0) \right)^{N-k} \tag{36}$$

$$\leq kN^k \left( 1 - P_X(x, 0) \right)^{N-k} \tag{37}$$

$$\leq kN^k e^{-(N-k)P_X(x, 0)} \tag{38}$$

Now let's consider the case when $\rho_{k,1} = 0$. We can write the term $\zeta_1$ in the form:

$$\zeta_1 = \psi(\tilde{k}_1) + \sum_{l=1}^{d} \left( \mathbf{1}_{\{\mathrm{pa}(X_l) \neq \emptyset\}} \log(n_{\mathrm{pa}(X_l)}^{(1)} + 1) - \log(n_{\mathrm{pa+}(X_l)}^{(1)} + 1) \right) + K_N$$

The term $K_N$ depends only on $N$ and the structure of the Bayesian model $\overline{\mathbb{P}}_X$ and is independent of the observed samples, and in general is equal to:

$$K_N = -\log C_{d_X} + \sum_{l=1}^{d}\left(\log C_{d_{\text{pa}+(X_l)}} - \log C_{d_{\text{pa}(X_l)}}\right) + \left(\sum_{l=1}^{d}\mathbf{1}_{\{\text{pa}(X_l)=\emptyset\}} - 1\right)\log N. \quad (39)$$

In which the terms $C_{d_S}$ indicate the volume of a unit ball in an $d_S$-dimensional space $S$. Since we are using $\ell_\infty$-norm in our algorithm and proofs, all $C_{d_S}$ terms will be equal to 1 and hence $K_N = \left(\sum_{l=1}^{d}\mathbf{1}_{\{\text{pa}(X_l)=\emptyset\}} - 1\right)\log N$ as it appeared in Algorithm 1.

Then for the case of $\rho_{k,1} = 0$ we can write:

$$\left|\mathbb{E}[\zeta_1|X=x,\rho_{k,1}=0] - \log f(x)\right| \quad (40)$$

$$= \left|\mathbb{E}\left[\psi(\tilde{k}_i) + \sum_{l=1}^{d}\left(\mathbf{1}_{\{\text{pa}(X_l)\neq\emptyset\}}\log(n^{(1)}_{\text{pa}(X_l)}+1) - \log(n^{(1)}_{\text{pa}+(X_l)}+1)\right)\right.\right.$$

$$\left.\left. + \left(\sum_{l=1}^{d}\mathbf{1}_{\{\text{pa}(X_l)=\emptyset\}} - 1\right)\log N \middle| X=x,\rho_{k,1}=0\right]\right.$$

$$\left. - \log P_X(x,0)\prod_{l=1}^{d}\frac{P_{\text{pa}(X_l)}\left(x_{\text{pa}(l)},0\right)}{P_{\text{pa}+(X_l)}\left(x_{\text{pa}+(l)},0\right)}\right| \quad (41)$$

$$\leq \left|\mathbb{E}[\psi(\tilde{k}_1)|X=x,\rho_{k,1}=0] - \log N P_X(x,0)\right| \quad (42)$$

$$+ \sum_{l=1}^{d}\left|\mathbb{E}[\log(n^{(1)}_{\text{pa}+(X_l)}+1)|X=x,\rho_{k,1}=0] - \log N P_{\text{pa}+(X_l)}\left(x_{\text{pa}+(l)},0\right)\right| \quad (43)$$

$$+ \sum_{l=1}^{d}\mathbf{1}_{\{\text{pa}(X_l)\neq\emptyset\}}\left|\mathbb{E}[\log(n^{(1)}_{\text{pa}(X_l)}+1)|X=x,\rho_{k,1}=0] - \log N P_{\text{pa}(X_l)}\left(x_{\text{pa}(l)},0\right)\right| \quad (44)$$

We notice that $\tilde{k}_1$ is the number of samples among $\{x^{(i)}\}_{i=1}^{N}$ such that $X_i = x$, where each sample is independently equal to $x$ with probability $P_X(x,0)$. Therefore the distribution of $\tilde{k}_1$ is $\text{Bino}\left(N, P_X(x,0)\right)$. Similarly, for any $S \subset X$, $n_{S,1}+1$ is the number of samples among $\{x^{(i)}\}_{i=1}^{N}$ such that $s^{(i)} = s$; i.e. projection of $x^{(i)}$ over $S$ is equal to the projection of $x$ over $S$. Thus $n_{S,1}+1 \sim \text{Bino}\left(N, P_S(s,0)\right)$. In addition to that, the event $\rho_{k,1}=0$ is equivalent to $\tilde{k}_1 \geq k$. Thus to upperbound term 42 and any of the individual terms inside the summations of terms 43 and 44 we propose the lemma below:

**Lemma 4.** *If $X$ is distributed as $Bino(N,p)$ and $m \geq 0$, then:*

$$\left|\mathbb{E}\left[\log(X+m)|X \geq k\right] - \log(Np)\right| \leq U(k,N,m,p) \quad (45)$$

*Where $U(k,N,m,p)$ is given by:*

$$U(k,N,m,p) = \max\left\{\left|\log\left(\frac{1+\frac{m}{Np}}{1-\exp\left(-2\frac{(Np-k)^2}{N}\right)}\right)\right|, \frac{1}{1-\exp\left(-2\frac{(Np-k)^2}{N}\right)}\frac{3}{2Np}\right\} \quad (46)$$

*Proof.* Please see Section F.3. □

**Remark.** *Since the Assumption 1.1 states that $k/N \to 0$ as $N \to \infty$, then $(Np-k)^2/N = N(p-k/N)^2 \to \infty$, and the upperbound $U(k,N,m,p)$ will converge to 0 as $N \to \infty$ for any $p$.*

From Lemma 4 we have:

$$\left| \mathbb{E}[\log(n_S^{(1)} + 1)|X = x, \rho_{k,1} = 0] - \log N P_S(s,0) \right| \tag{47}$$

$$= \left| \mathbb{E}[\log(n_S^{(1)} + 1)|X = x, n_S^{(1)} + 1 \geq k] - \log N P_S(s,0) \right| \leq U\Big(k, N, 0, P_S(s,0)\Big) \tag{48}$$

For any of the individual terms inside the summations of terms 43 and 44. For the term 42 we notice that $|\psi(\tilde{k}_1) - \log(\tilde{k}_1)| \leq 1/\tilde{k}_1 \leq 1/k$ [56]. Thus this term can be bounded similarly by $U\left(k, N, 0, P_X(x,0)\right) + 1/k$. By combining all these bounds, we obtain:

$$\left| \mathbb{E}[\zeta_1 | X = x, \rho_{k,1} = 0] - \log f(x) \right| \tag{49}$$

$$\leq \sum_{l=1}^{d} \left( U\Big(k, N, 0, P_{\text{pa}(X_l)}(x_{\text{pa}(l)}, 0)\Big) + U\Big(k, N, 0, P_{\text{pa+}(X_l)}(x_{\text{pa+}(l)}, 0)\Big) \right)$$

$$+ U\Big(k, N, 0, P_X(x,0)\Big) + \frac{1}{k} \tag{50}$$

Assumption 1.2 implies that discrete probabilities and hence $U\Big(k, N, 0, p\Big)$ terms are bounded; i.e. there exists a $\hat{p}$ such that for any $x \in \Omega_2$:

$$\begin{cases} U\Big(k, N, 0, P_X(x,0)\Big) \leq U\Big(k, N, 0, \hat{p}\Big) & \\ U\Big(k, N, 0, P_{\text{pa}(X_l)}(x_{\text{pa}(l)}, 0)\Big) \leq U\Big(k, N, 0, \hat{p}\Big) & \text{for } l = 1, \ldots, d \\ U\Big(k, N, 0, P_{\text{pa+}(X_l)}(x_{\text{pa+}(l)}, 0), 0)\Big) \leq U\Big(k, N, 0, \hat{p}\Big) & \text{for } l = 1, \ldots, d \end{cases} \tag{51}$$

Therefore:

$$\left| \mathbb{E}[\zeta_1 | X = x, \rho_{k,1} = 0] - \log f(x) \right| \leq (2d+1) U\Big(k, N, 0, \hat{p}\Big) + \frac{1}{k} \tag{52}$$

If we combine it with the case of $\rho_{k,i} > 0$, we obtain that:

$$\left| \mathbb{E}[\zeta_1 | X = x] - \log f(x) \right| \tag{53}$$

$$\leq \left| \mathbb{E}[\zeta_1 | X = x, \rho_{k,1} > 0] - \log f(x) \right| \times \mathbb{P}(\rho_{k,1} > 0)$$

$$+ \left| \mathbb{E}[\zeta_1 | X = x, \rho_{k,1} = 0] - \log f(x) \right| \times \mathbb{P}(\rho_{k,1} = 0) \tag{54}$$

$$\leq \left( (2d+1) \log N + |\log f(x)| \right) k N^k e^{-(N-k)P_X(x,0)} + (2d+1) U\Big(k, N, 0, \hat{p}\Big) + \frac{1}{k} \tag{55}$$

Where we used the fact that $|\zeta_1| \leq (2d+1) \log N$. Integrating over $\Omega_2$, we can write:

$$\int_{\Omega_2} \left| \mathbb{E}[\zeta_1 | X = x] - \log f(x) \right| d\mathbb{P}_X \tag{56}$$

$$\leq \left( (2d+1) \log N + \int_{\Omega_2} |\log f(x)| d\mathbb{P}_X \right) k N^k e^{-(N-k)\inf_{x \in \Omega_2} P_X(x,0)} \tag{57}$$

$$+ (2d+1) U\Big(k, N, 0, \hat{p}\Big) + \frac{1}{k} \tag{58}$$

By Assumption 1.1, $k$ goes to infinity as $N$ goes to infinity, so $1/k$ vanishes as $N$ grows boundlessly. The term $U\Big(k, N, 0, \hat{p}\Big)$ will also converge to zero as we saw. From assumption 1.3, $\int_{\Omega_2} |\log f(x)| d\mathbb{P}_X < +\infty$. Thus the first term also converges to 0 as $N \to \infty$. Thus:

$$\lim_{N \to \infty} \int_{\Omega_2} \left| \mathbb{E}[\zeta_1 | X = x] - \log f(x) \right| d\mathbb{P}_X = 0 \tag{59}$$

**For $x \in \Omega_3$:**

In this case, $P_X(x,r)$ is a monotonic function of $r$ such that $P_X(x,0) = 0$ and $\lim_{r \to \infty} P_X(x,r) = 1$. Hence we can view $\log P_X(x,r) \prod_{l=1}^{d} \frac{P_{\text{pa}(X_l)}\left(x_{\text{pa}(l)},r\right)}{P_{\text{pa+}(X_l)}\left(x_{\text{pa+}(l)},r\right)}$ as a function of $P_X(x,r)$ and it converges to $\log f(x)$ as $P_X(x,r) \to 0$, for almost every $x$. Since $\mathbb{P}_X(\Omega_3) \le 1 < \infty$ and we know $\int_{\Omega_3} |\log f(x)| \, d\mathbb{P}_X < \infty$, By Egorov's theorem, for any $\epsilon > 0$, there exists a subset $E \subseteq \Omega_3$ with $\mathbb{P}_X(E) < \epsilon$ and $\int_E |\log f(x)| \, d\mathbb{P}_X < \epsilon$, such that the term $\log P_X(x,r) \prod_{l=1}^{d} \frac{P_{\text{pa}(X_l)}\left(x_{\text{pa}(l)},r\right)}{P_{\text{pa+}(X_l)}\left(x_{\text{pa+}(l)},r\right)}$ converges uniformly on $\Omega_3 \setminus E$ as $P_X(x,r) \to 0$. Now let's assume $\epsilon_N$ is a sequence converging to 0. Consequently, The corresponding sets $E_N$ will also create a sequence. For any fixed $N$ and for $x \in E_N$, we notice that $|\zeta_1| \le (2d+1) \log N$, so we have:

$$\int_{E_N} \left| \mathbb{E}[\zeta_1 | X = x] - \log f(x) \right| d\mathbb{P}_X \tag{60}$$

$$\le \int_{E_N} \left( (2d+1) \log N + |\log f(x)| \right) d\mathbb{P}_X < \epsilon_N \left( (2d+1) \log N + 1 \right) \tag{61}$$

By choosing $\epsilon_N$ such that $\epsilon_N \log N \to 0$ as $N \to \infty$ (For example $\epsilon_N = 1/N$), we will have $\lim_{N \to \infty} \int_E |\mathbb{E}[\zeta_1 | X = x] - \log f(x)| \, d\mathbb{P}_X = 0$.

For any $x \in \Omega_3 \setminus E_N$, since $P_X(x,0) = 0$, we know that $\mathbb{P}(\rho_{k,1} = 0 | X = x) = 0$, so $\tilde{k}_1 = k$ with probability 1. Conditioning on $\rho_{k,1} = r > 0$, the term $|\mathbb{E}[\zeta_1 | X = x] - \log f(x)|$ can be decomposed as:

$$\left| \mathbb{E}[\zeta_1 | X = x] - \log f(x) \right| \tag{62}$$

$$= \left| \int_{r=0}^{\infty} \left( \mathbb{E}[\zeta_1 | X = x, \rho_{k,1} = r] - \log f(x) \right) dF_{\rho_{k,1}}(r) \right| \tag{63}$$

$$\le \left| \int_{r=0}^{\infty} \left( \log P_X(x,r) \prod_{l=1}^{d} \frac{P_{\text{pa}(X_l)}\left(x_{\text{pa}(l)},r\right)}{P_{\text{pa+}(X_l)}\left(x_{\text{pa+}(l)},r\right)} - \log f(x) \right) dF_{\rho_{k,1}}(r) \right| \tag{64}$$

$$+ \left| \int_{r=0}^{\infty} \left( \psi(k) - \log N - \log P_X(x,r) \right) dF_{\rho_{k,1}}(r) \right| \tag{65}$$

$$+ \sum_{l=1}^{d} \left| \int_{r=0}^{\infty} \left( \mathbb{E}\left[ \log(n_{\text{pa+}(X_l)}^{(1)} + 1) | (X, \rho_{k,1}) = (x,r) \right] \right. \right.$$

$$\left. \left. - \log N P_{\text{pa+}(X_l)}(x_{\text{pa+}(l)},r) \right) dF_{\rho_{k,1}}(r) \right| \tag{66}$$

$$+ \sum_{l=1}^{d} \mathbf{1}_{\{\text{pa}(X_l) \ne \emptyset\}} \left| \int_{r=0}^{\infty} \left( \mathbb{E}\left[ \log(n_{\text{pa}(X_l)}^{(1)} + 1) | (X, \rho_{k,1}) = (x,r) \right] \right. \right.$$

$$\left. \left. - \log N P_{\text{pa}(X_l)}(x_{\text{pa}(l)},r) \right) dF_{\rho_{k,1}}(r) \right| \tag{67}$$

In which $F_{\rho_{k,1}}(r)$ is the CDF of the $k$-nearest neighbor distance $\rho_{k,1}$ given $X = x$. The derivative of this CDF with respect to $P_X(x,r)$ is given by:

$$\frac{dF_{\rho_{k,1}}(r)}{dP_X(x,r)} = \frac{(N-1)!}{(k-1)!(N-k-1)!} P_X(x,r)^{k-1} \left( 1 - P_X(x,r) \right)^{N-k-1} \tag{68}$$

*Upper bound for the term (64) :*

Since $P_X(x,r) \prod_{l=1}^{d} \frac{P_{\text{pa}(X_l)}\left(x_{\text{pa}(l)},r\right)}{P_{\text{pa+}(X_l)}\left(x_{\text{pa+}(l)},r\right)}$ converges to $f(x)$ uniformly over $\Omega_3 \setminus E$ as $P_X(x,r) \to 0$, So for any $\delta_N$ and any $x \in \Omega_3 \setminus E$, there exists $r_1$ such that for any $r < r_1$:

$$\left| \log P_X(x,r) \prod_{l=1}^{d} \frac{P_{\text{pa}(X_l)}\left(x_{\text{pa}(l)},r\right)}{P_{\text{pa+}(X_l)}\left(x_{\text{pa+}(l)},r\right)} - \log f(x) \right| < \delta_N \tag{69}$$

Let $r_2$ be the value of $r$ such that $P_X(x, r_2) = 4k \log N/N$, and take $r_N = \min\{r_1, r_2\}$. Thus $r_N$ depends on $x$, but $\delta_N$ does not depend on $x$ and $\lim_{N\to\infty} \delta_N = 0$. Therefore, (64) can be upper bounded as:

$$\left| \int_{r=0}^{\infty} \left( \log P_X(x, r) \prod_{l=1}^{d} \frac{P_{\mathrm{pa}(X_l)}\left(x_{\mathrm{pa}(l)}, r\right)}{P_{\mathrm{pa}+(X_l)}\left(x_{\mathrm{pa}+(l)}, r\right)} - \log f(x) \right) dF_{\rho_{k,1}}(r) \right| \tag{70}$$

$$\leq \int_{r=0}^{r_N} \left| \log P_X(x, r) \prod_{l=1}^{d} \frac{P_{\mathrm{pa}(X_l)}\left(x_{\mathrm{pa}(l)}, r\right)}{P_{\mathrm{pa}+(X_l)}\left(x_{\mathrm{pa}+(l)}, r\right)} - \log f(x) \right| dF_{\rho_{k,1}}(r)$$

$$+ \int_{r=r_N}^{\infty} \left| \log P_X(x, r) \prod_{l=1}^{d} \frac{P_{\mathrm{pa}(X_l)}\left(x_{\mathrm{pa}(l)}, r\right)}{P_{\mathrm{pa}+(X_l)}\left(x_{\mathrm{pa}+(l)}, r\right)} - \log f(x) \right| dF_{\rho_{k,1}}(r) \tag{71}$$

$$\leq \delta_N \mathbb{P}\left(\rho_{k,1} \leq r_N | X = x\right)$$

$$+ \left( \sup_{r \geq r_N} \left| \log P_X(x, r) \prod_{l=1}^{d} \frac{P_{\mathrm{pa}(X_l)}\left(x_{\mathrm{pa}(l)}, r\right)}{P_{\mathrm{pa}+(X_l)}\left(x_{\mathrm{pa}+(l)}, r\right)} - \log f(x) \right| \right) \mathbb{P}\left(\rho_{k,1} \geq r_N | X = x\right) \tag{72}$$

First, $\mathbb{P}\left(\rho_{k,1} \leq r_N | X = x\right)$ is smaller than 1. Secondly, since $P_X(x, r) \geq 4k \log N/N > 1/N$ for $r \geq r_N$, so we have $|\log P_X(x, r)| \leq \log N$. The same bounds apply for any $|P_S(s, r)|$ as well, as $P_S(s, r) \geq P_X(x, r)$. Thus by triangle inequality, the supremum is upper-bounded by $(2d + 1) \log N + |\log f(x)|$. Finally, the probability $\mathbb{P}\left(\rho_{k,1} \geq r_N | X = x\right)$ is upper bounded by:

$$\mathbb{P}\left(\rho_{k,1} \geq r_N | X = x\right) \tag{73}$$

$$= \sum_{m=0}^{k-1} \binom{N-1}{m} P_X(x, r_N)^m \left(1 - P_X(x, r_N)\right)^{N-1-m} \tag{74}$$

$$\leq \sum_{m=0}^{k-1} N^m \left(1 - P_X(x, r_N)\right)^{N-k} \tag{75}$$

$$= kN^k \left(1 - \frac{4k \log N}{N}\right)^{N/2} \tag{76}$$

$$\leq kN^k e^{-2k \log N} = \frac{k}{N^k} \tag{77}$$

for $N$ large enough such that $N - k > N/2$. Therefore 64 is upperbounded by:

$$\left| \int_{r=0}^{\infty} \left( \log P_X(x, r) \prod_{l=1}^{d} \frac{P_{\mathrm{pa}(X_l)}\left(x_{\mathrm{pa}(l)}, r\right)}{P_{\mathrm{pa}+(X_l)}\left(x_{\mathrm{pa}+(l)}, r\right)} - \log f(x) \right) dF_{\rho_{k,1}}(r) \right| \tag{78}$$

$$\leq \delta_N + k \frac{(2d+1) \log N + |\log f(x)|}{N^k} \tag{79}$$

*Upper bound for the term (65) :*

We have:

$$\int_{r=0}^{\infty} \left(\psi(k) - \log N - \log P_X(x,r)\right) dF_{\rho_{k,1}}(r) \tag{80}$$

$$= \psi(k) - \log N - \frac{(N-1)!}{(k-1)!(N-k-1)!}$$

$$\times \int_{r=0}^{\infty} \left(P_X(x,r)^{k-1}(1 - P_X(x,r))^{N-k-1} \log P_X(x,r)\right) dP_X(x,r) \tag{81}$$

$$= \psi(k) - \log N - \frac{(N-1)!}{(k-1)!(N-k-1)!} \int_{t=0}^{1} \left(t^{k-1}(1-t)^{N-k-1} \log t\right) dt \tag{82}$$

$$= \psi(k) - \log N - (\psi(k) - \psi(N)) \tag{83}$$

$$= \psi(N) - \log(N) \tag{84}$$

We notice that for any $N$ we have $\psi(N) < \log N$ and $\lim_{N \to \infty} (\psi(N) - \log N) = 0$.

*Upper bound for the individual terms of (66) and (67) :*
The upper bound for each of these terms follows the same logic, so we do the proof for an arbitrary subset $S \in \{pa(X_l)\}_{l=1}^{d} \cup \{pa+(X_l)\}_{l=1}^{d}$.

The distributions of $n_S^{(1)}$ for all $S \in \{pa(X_l)\}_{l=1}^{d} \cup \{pa+(X_l)\}_{l=1}^{d}$ are given by the lemma below:

**Lemma 5.** *Given $X = x$ and $\rho_{k,1} = r > 0$ and $s$ being the projection of $x$ over $S$, the term $n_S^{(1)} - k$ is distributed as $Bino\left(N - k - 1, \frac{P_S(s,r) - P_X(x,r)}{1 - P_X(x,r)}\right)$.*

*Proof.* Please see the proof of Lemma B.2 in [51]. $\qquad\square$

The bound on $\mathbb{E}[\log(n_S^{(1)} + 1)|X = x, \rho_{k,1} = r]$ is given by the Lemma B.3 in [51] which we restate here:

**Lemma 6.** *If $X$ is distributed as $Bino(N, p)$, then $|\mathbb{E}[\log(X + k)] - \log(Np + k)| \leq C/(Np + k)$ for some constant $C$.*

*Proof.* Please see the proof of Lemma B.3 in [51]. $\qquad\square$

Thus we can write:

$$\left| \int_{r=0}^{\infty} \left( \mathbb{E}\left[\log(n_S^{(1)} + 1)|(X, \rho_{k,1}) = (x,r)\right] - \log N P_S(s,r) \right) dF_{\rho_{k,1}}(r) \right| \tag{85}$$

$$\leq \left| \int_{r=0}^{\infty} \left( \mathbb{E}[\log(n_S^{(1)} + 1)|(X, \rho_{k,1}) = (x,r)] \right. \right.$$

$$\left. \left. - \log\left((N-k-1)\frac{P_S(s,r) - P_X(x,r)}{1 - P_X(x,r)} + k + 1\right) \right) dF_{\rho_{k,1}}(r) \right| \tag{86}$$

$$+ \left| \int_{r=0}^{\infty} \left( \log \frac{(N-k-1)\frac{P_S(s,r) - P_X(x,r)}{1 - P_X(x,r)} + k + 1}{N P_S(s,r)} \right) dF_{\rho_{k,1}}(r) \right| \tag{87}$$

$$\leq \int_{r=0}^{\infty} \left| \left( \mathbb{E}\left[\log(n_S^{(1)} + 1)|(X, \rho_{k,1}) = (x,r)\right] \right. \right.$$

$$\left. - \log\left((N-k-1)\frac{P_S(s,r) - P_X(x,r)}{1 - P_X(x,r)} + k + 1\right) \right) \left| dF_{\rho_{k,1}}(r) \right. \tag{88}$$

$$+ \left| \mathbb{E}_r \left[ \log \frac{N(P_S(s,r) - P_X(x,r)) + (k+1)(1 - P_S(s,r))}{N P_S(s,r)(1 - P_X(x,r))} \right] \right| \tag{89}$$

Where $\mathbb{E}_r$ denotes the expectation over the distribution $F_{\rho_{k,1}}$. By the Lemma B.3 in [51], the term in 88 is upper bounded by:

$$\int_{r=0}^{\infty} \left| \left( \mathbb{E}\left[ \log(n_S^{(1)} + 1) | (X, \rho_{k,1}) = (x, r) \right] \right. \right.$$

$$\left. \left. - \log\left( (N - k - 1) \frac{P_S(s, r) - P_X(x, r)}{1 - P_X(x, r)} + k + 1 \right) \right) \right| dF_{\rho_{k,1}}(r) \quad (90)$$

$$\leq \int_{r=0}^{\infty} \frac{C}{(N - k - 1) \frac{P_S(s,r) - P_X(x,r)}{1 - P_X(x,r)} + k + 1} dF_{\rho_{k,1}}(r) \quad (91)$$

$$\leq \int_{r=0}^{\infty} \frac{C}{k+1} dF_{\rho_{k,1}}(r) = \frac{C}{k+1} \quad (92)$$

The last inequality follows from the fact that $P_S(s, r) > P_X(x, r)$ . For (89), using the fact that $\log(x/y) \leq (x - y)/y$ and Cauchy-Schwarz inequality, we have the following:

$$\mathbb{E}_r \left[ \log \frac{N(P_S(s,r) - P_X(x,r)) + (k+1)(1 - P_S(s,r))}{N P_S(s,r)(1 - P_X(x,r))} \right] \quad (93)$$

$$\leq \mathbb{E}_r \left[ \frac{N(P_S(s,r) - P_X(x,r)) + (k+1)(1 - P_S(s,r))}{N P_S(s,r)(1 - P_X(x,r))} - 1 \right] \quad (94)$$

$$= \mathbb{E}_r \left[ \frac{(k + 1 - N P_X(x,r))(1 - P_S(s,r))}{N P_S(s,r)(1 - P_X(x,r))} \right] \quad (95)$$

$$\leq \sqrt{ \mathbb{E}_r \left[ \left( \frac{k + 1 - N P_X(x,r)}{N P_X(x,r)} \right)^2 \right] \mathbb{E}_r \left[ \left( \frac{P_X(x,r)(1 - P_S(s,r))}{P_S(s,r)(1 - P_X(x,r))} \right)^2 \right] } \quad (96)$$

Note that $P_S(s, r) \geq P_X(x, r)$ for all $r$, so the second expectation term is always smaller than or equal to 1. For the first expectation, let $t = P_X(x, r)$ then we have:

$$\mathbb{E}_r \left[ \left( \frac{k + 1 - N P_X(x,r)}{N P_X(x,r)} \right)^2 \right] \quad (97)$$

$$= \int_{r=0}^{\infty} \left( \frac{k + 1 - N P_X(x,r)}{N P_X(x,r)} \right)^2 dF_{\rho_{k,1}}(r) \quad (98)$$

$$= \frac{(N-1)!}{(k-1)!(N-k-1)!} \int_{t=0}^{1} \frac{(k + 1 - Nt)^2}{N^2 t^2} t^{k-1}(1-t)^{N-k-1} dt \quad (99)$$

$$= \frac{(N-1)(N-2)(k+1)^2}{N^2(k-1)(k-2)} - \frac{2(N-1)(k+1)}{N(k-1)} + 1 \quad (100)$$

This term is upper bounded by $C_1(1/N + 1/k)$ for some constant $C_1$ for $N$ and $k$ large enough. Thus:

$$\mathbb{E}_r \left[ \log \frac{N(P_S(s,r) - P_X(x,r)) + (k+1)(1 - P_S(s,r))}{N P_S(s,r)(1 - P_X(x,r))} \right] \leq \sqrt{C_1 \left( \frac{1}{N} + \frac{1}{k} \right)} \quad (101)$$

Similarly, by using the fact that $\log(x/y) > (x - y)/x$ and Cauchy-Schwarz inequality again, we conclude that there are some constant $C_2 > 0$ such that:

$$\mathbb{E}_r \left[ \log \frac{N(P_S(s,r) - P_X(x,r)) + (k+1)(1 - P_S(s,r))}{N P_S(s,r)(1 - P_X(x,r))} \right] \geq -\sqrt{C_2 \left( \frac{1}{N} + \frac{1}{k} \right)} \quad (102)$$

Therefore by combining all these bounds we obtain

$$\left| \int_{r=0}^{\infty} \left( \mathbb{E}\left[ \log(n_S^{(1)} + 1) | (X, \rho_{k,1}) = (x, r) \right] - \log N P_S(s, r) \right) dF_{\rho_{k,1}}(r) \right| \quad (103)$$

$$\leq \frac{C}{k+1} + \sqrt{C' \left( \frac{1}{k} + \frac{1}{N} \right)} \quad (104)$$

where $C' = \max\{C_1, C_2\}$.

Now putting all the bounds for $\Omega_3 \setminus E_N$, we have:

$$\int_{\Omega_3 \setminus E_N} \left| \mathbb{E}\left[\zeta_1 | X = x\right] - \log f(x) \right| d\mathbb{P}_X \tag{105}$$

$$\leq \quad \delta_N + k \frac{(2d+1) \log N + \int_{\mathcal{X}} |\log f(x)| \, d\mathbb{P}_X}{N^k}$$

$$+ \psi(N) - \log(N) + 2d \left( \frac{C}{k+1} + \sqrt{C'(\frac{1}{k} + \frac{1}{N})} \right) \tag{106}$$

From the assumptions 1, $k$ increases as $N \to \infty$, and $\int_{\mathcal{X}} |\log f(x)| \, d\mathbb{P}_X < \infty$. Therefore the whole upperbound vanishes as $N$ goes to infinity. Thus for the entire set $\Omega_3$ we have:

$$\lim_{N \to \infty} \int_{\Omega_3} \left| \mathbb{E}[\zeta_1 | X = x)] - \log f(x) \right| d\mathbb{P}_X = 0 \tag{107}$$

### F.2 Proof of Lemma 3

This proof is based on the Lebesgue-Besicovitch differentiation theorem (For example look at Theorem 1.32 from [57]), stated as:

**Theorem 7.** *Let $\mu$ be a Radon measure on $\mathbb{R}^d$. For $f \in L^1_{loc}(\mu)$,*

$$\lim_{r \to 0} \frac{1}{\mu\left(B_r(x)\right)} \int_{B_r(x)} f d\mu = f(x) \tag{108}$$

*for $\mu$-a.e. $x$.*

Let $f = \frac{d\mathbb{P}_X}{d\overline{\mathbb{P}}_X}$ and $\mu = \overline{\mathbb{P}}_X$. Since $\mu$ is a probability measure, it is a Radon measure on Euclidean space. Also, since $\int_{\mathcal{X}} |f| d\mu = 1$, so it's globally and therefore locally integrable with respect to $\mu$. Thus the conditions of the Theorem 7 are satisfied, and we can write:

$$\lim_{r \to 0} P_X(x, r) \prod_{l=1}^{d} \frac{P_{\text{pa}(X_l)}\left(x_{\text{pa}(l)}, r\right)}{P_{\text{pa}+(X_l)}\left(x_{\text{pa}+(l)}, r\right)} \tag{109}$$

$$= \lim_{r \to 0} \frac{\mathbb{P}\left\{B_r(x)\right\}}{\prod_{l=1}^{d} \mathbb{P}_{X_l | \text{pa}(X_l)}\left\{B_r(x_l) \middle| B_r\left(x_{\text{pa}(l)}\right)\right\}} \tag{110}$$

$$= \lim_{r \to 0} \frac{\mathbb{P}\left\{B_r(x)\right\}}{\overline{\mathbb{P}}_X\left\{B_r(x)\right\}} \tag{111}$$

$$= \lim_{r \to 0} \frac{1}{\overline{\mathbb{P}}_X\left\{B_r(x)\right\}} \int_{B_r(x)} \frac{d\mathbb{P}_X}{d\overline{\mathbb{P}}_X} d\overline{\mathbb{P}}_X \tag{112}$$

$$= \frac{d\mathbb{P}_X}{d\overline{\mathbb{P}}_X}(x) \tag{113}$$

## F.3 Proof of Lemma 4

First, we upperbound $\mathbb{E}\left[\log(X+m)|X \geq k\right] - \log(Np)$. We can see that:

$$\mathbb{E}\left[X+m|X \geq k\right] \tag{114}$$

$$= \frac{1}{\mathbb{P}\left(X \geq k\right)} \sum_{i=k}^{N}(i+m)\left(\begin{array}{c}N\\i\end{array}\right)p^i(1-p)^{N-i} \tag{115}$$

$$\leq \frac{1}{1-\exp\left(-2\frac{(Np-k)^2}{N}\right)} \sum_{i=k}^{N}(i+m)\left(\begin{array}{c}N\\i\end{array}\right)p^i(1-p)^{N-i} \tag{116}$$

$$\leq \frac{1}{1-\exp\left(-2\frac{(Np-k)^2}{N}\right)} \sum_{i=1}^{N}(i+m)\left(\begin{array}{c}N\\i\end{array}\right)p^i(1-p)^{N-i} \tag{117}$$

$$= \frac{1}{1-\exp\left(-2\frac{(Np-k)^2}{N}\right)}\left(\mathbb{E}\left[X\right]+m\right) = \frac{Np+m}{1-\exp\left(-2\frac{(Np-k)^2}{N}\right)} \tag{118}$$

In which we used the Hoeffding's inequality. We know that $\mathbb{E}\left[\log(X+m)|X \geq k\right] \leq \log\left(\mathbb{E}\left[X+m|X \geq k\right]\right)$, therefore:

$$\mathbb{E}\left[\log(X)|X \geq k\right] - \log(Np) \leq \log\left(\frac{1+\frac{m}{Np}}{1-\exp\left(-2\frac{(Np-k)^2}{N}\right)}\right) \tag{119}$$

Second, to give an upper bound over $\log(Np) - \mathbb{E}\left[\log(X+m)|X \geq k\right]$, we first notice that:

$$\log(Np) - \mathbb{E}\left[\log(X+m)|X \geq k\right] \leq \log(Np) - \mathbb{E}\left[\log(X)|X \geq k\right] \tag{120}$$

Then we upperbound $\log(Np) - \mathbb{E}\left[\log(X)|X \geq k\right]$ by applying Taylor's theorem around $x_0 = Np$, where there exists $\zeta$ between $x$ and $x_0$ such that:

$$\log(x) = \log(Np) + \frac{x-Np}{Np} - \frac{(x-Np)^2}{2\zeta^2} \tag{121}$$

since $\zeta \geq \min\{x, x_0\} = \min\{x, Np\}$, we have:

$$-\log(x) + \log(Np) + \frac{x-Np}{Np} = \frac{(x-Np)^2}{2\zeta^2}$$

$$\leq \max\left\{\frac{(x-Np)^2}{2x^2}, \frac{(x-NP)^2}{2(Np)^2}\right\} \leq \frac{(x-Np)^2}{2x^2} + \frac{(x-Np)^2}{2(Np)^2} \tag{122}$$

Now taking the conditional expectations from both sides, we have:

$$-\mathbb{E}\left[\log(X)|X \geq k\right] + \log(Np) + \frac{\mathbb{E}\left[X|X \geq k\right] - Np}{Np}$$

$$\leq \mathbb{E}\left[\frac{(X-Np)^2}{2X^2}\bigg|X \geq k\right] + \frac{\mathbb{E}\left[(X-Np)^2\big|X \geq k\right]}{2(Np)^2} \tag{123}$$

First, we notice that $\mathbb{E}\left[X|X \geq k\right] \geq \mathbb{E}\left[X\right] = Np$.

Second, $\mathbb{E}\left[(X-Np)^2\big|X \geq k\right] \leq \frac{1}{1-\exp\left(-2\frac{(Np-k)^2}{N}\right)}\text{Var}\left[X\right] = \frac{Np(1-p)}{1-\exp\left(-2\frac{(Np-k)^2}{N}\right)}.$

Thus we can write:

$$-\mathbb{E}\left[\log(X)|X \geq k\right] + \log(Np) \leq \frac{Np(1-p)}{1-\exp\left(-2\frac{(Np-k)^2}{N}\right)}\frac{1}{2(Np)^2} + \mathbb{E}\left[\frac{(X-Np)^2}{2X^2}\bigg|X \geq k\right] \tag{124}$$

To deal with the term $\mathbb{E}\left[\frac{(X-Np)^2}{2X^2}\Big| X \geq k\right]$, we have:

$$\mathbb{E}\left[\frac{(X-Np)^2}{2X^2}\Big| X \geq k\right] \tag{125}$$

$$\leq \frac{1}{1-\exp\left(-2\frac{(Np-k)^2}{N}\right)}\sum_{i=k}^{N}\frac{(i-Np)^2}{2i^2}\left(\begin{array}{c}N\\i\end{array}\right)p^i(1-p)^{N-i} \tag{126}$$

$$\leq \frac{1}{1-\exp\left(-2\frac{(Np-k)^2}{N}\right)}\sum_{i=k}^{N}\frac{(i-Np)^2}{(i+1)(i+2)}\left(\begin{array}{c}N\\i\end{array}\right)p^i(1-p)^{N-i} \tag{127}$$

$$= \frac{1}{1-\exp\left(-2\frac{(Np-k)^2}{N}\right)}\sum_{i=k}^{N}\frac{(i-Np)^2}{(N+1)(N+2)p^2}\left(\begin{array}{c}N+2\\i+2\end{array}\right)p^{2+i}(1-p)^{N-i} \tag{128}$$

$$\leq \frac{1}{1-\exp\left(-2\frac{(Np-k)^2}{N}\right)}\frac{1}{(N+1)(N+2)p^2}\mathbb{E}_{Y\sim\text{Bino}(N+2,p)}\left[(Y-Np)^2\right] \tag{129}$$

$$= \frac{1}{1-\exp\left(-2\frac{(Np-k)^2}{N}\right)}\frac{(N+2)p(1-p)+4p^2}{(N+1)(N+2)p^2} \tag{130}$$

$$\leq \frac{1}{1-\exp\left(-2\frac{(Np-k)^2}{N}\right)}\frac{(N+2)p}{(N+1)(N+2)p^2} \leq \frac{1}{1-\exp\left(-2\frac{(Np-k)^2}{N}\right)}\frac{1}{Np} \tag{131}$$

In which we used the fact that $2i^2 \geq (i+1)(i+2)$ for $i \geq 4$, and $(N+2)p \geq 4p$ for $N \geq 2$. Plugging it into Equation 124, we have:

$$-\mathbb{E}\left[\log(X)|X \geq k\right] + \log(Np) \leq \frac{1}{1-\exp\left(-2\frac{(Np-k)^2}{N}\right)}\frac{3}{2Np} \tag{132}$$

And the desired result is yielded.

### F.4 Proof of Theorem 2

For this part, we also follow the same procedure as followed for the Theorem 2 in [51] using Efron-Stein inequality. Suppose $\widehat{\text{GDM}}^{(N)}(X,\mathcal{G})$ is the estimate based on the original samples $x^{(1)}, x^{(2)}, \ldots, x^{(N)}$. For the usage of Efron-Stein inequality, we suppose that there is another set of $n$ i.i.d samples drawn from $P_X$ denoted by $x'^{(1)}, x'^{(2)}, \ldots, x'^{(n)}$. Let $\widehat{\text{GDM}}^{(N)}(X^{(j)},\mathcal{G})$ be the estimate based on the original samples' set in which the $j$th sample is replaced by another i.i.d sample $x'^{(j)}$ taken from the second set, i.e. $\widehat{\text{GDM}}^{(N)}(X^{(j)},\mathcal{G})$ is the estimate based on $x^{(1)}, \ldots, x^{(j-1)}, x'^{(j)}, x^{(j+1)}, \ldots, x^{(N)}$. Then the Efron-Stein inequality states that:

$$\text{Var}\left[\widehat{\text{GDM}}^{(N)}(X,\mathcal{G})\right] \leq \frac{1}{2}\sum_{j=1}^{N}\mathbb{E}\left[\left(\widehat{\text{GDM}}^{(N)}(X,\mathcal{G}) - \widehat{\text{GDM}}^{(N)}(X^{(j)},\mathcal{G})\right)^2\right] \tag{133}$$

Now we will find an upper bound for the difference $\left|\widehat{\text{GDM}}^{(N)}(X,\mathcal{G}) - \widehat{\text{GDM}}^{(N)}(X^{(j)},\mathcal{G})\right|$ for a given index $j$ by considering the worst case scenario. Let $\widehat{\text{GDM}}^{(N)}(X_{\backslash j},\mathcal{G})$ be the estimate based on the original samples' set in which the $j$th sample is removed, i.e. $x^{(1)}, \ldots, x^{(j-1)}, x^{j+1}, \ldots, x^{(N)}$.

Then by triangle inequality, we have:

$$\sup_{x^{(1)},\ldots,x^{(N)},x'^{(j)}} \left| \widehat{\mathbb{GDM}}^{(N)}(X,\mathcal{G}) - \widehat{\mathbb{GDM}}^{(N)}(X^{(j)},\mathcal{G}) \right|$$

$$\leq \sup_{x^{(1)},\ldots,x^{(N)},x'^{(j)}} \left( \left| \widehat{\mathbb{GDM}}^{(N)}(X,\mathcal{G}) - \widehat{\mathbb{GDM}}^{(N)}(X_{\backslash j},\mathcal{G}) \right| \right. \tag{134}$$

$$\left. + \left| \widehat{\mathbb{GDM}}^{(N)}(X_{\backslash j},\mathcal{G}) - \widehat{\mathbb{GDM}}^{(N)}(X^{(j)},\mathcal{G}) \right| \right)$$

$$\leq \sup_{x^{(1)},\ldots,x^{(N)}} \left| \widehat{\mathbb{GDM}}^{(N)}(X,\mathcal{G}) - \widehat{\mathbb{GDM}}^{(N)}(X_{\backslash j},\mathcal{G}) \right| \tag{135}$$

$$+ \sup_{x^{(1)},\ldots,x^{(j-1)},x'^{(j)},x^{(j+1)},\ldots,x^{(N)}} \left| \widehat{\mathbb{GDM}}^{(N)}(X_{\backslash j},\mathcal{G}) - \widehat{\mathbb{GDM}}^{(N)}(X^{(j)},\mathcal{G}) \right|$$

$$= 2 \sup_{x^{(1)},\ldots,x^{(N)}} \left| \widehat{\mathbb{GDM}}^{(N)}(X,\mathcal{G}) - \widehat{\mathbb{GDM}}^{(N)}(X_{\backslash j},\mathcal{G}) \right| \tag{136}$$

Remember that:

$$\widehat{\mathbb{GDM}}^{(N)}(X,\mathcal{G})$$

$$= \frac{1}{N} \sum_{i=1}^{N} \zeta_i(X) \tag{137}$$

$$= \frac{1}{N} \sum_{i=1}^{N} \left( \psi(\tilde{k}_i) + \sum_{l=1}^{d} \left( \mathbf{1}_{\{\mathrm{pa}(X_l)\neq\emptyset\}} \log(n_{\mathrm{pa}(X_l)}^{(i)} + 1) - \log(n_{\mathrm{pa+}(X_l)}^{(i)} + 1) \right) \right.$$

$$\left. + \left( \sum_{l=1}^{d} \mathbf{1}_{\{\mathrm{pa}(X_l)=\emptyset\}} - 1 \right) \log N \right) \tag{138}$$

Thus we can write:

$$\sup_{x^{(1)},\ldots,x^{(N)}} \left| \widehat{\mathbb{GDM}}^{(N)}(X,\mathcal{G}) - \widehat{\mathbb{GDM}}^{(N)}(X^{(j)},\mathcal{G}) \right| \leq \frac{2}{N} \sup_{x^{(1)},\ldots,x^{(N)}} \sum_{i=1}^{N} \left| \zeta_i(X) - \zeta_i(X_{\backslash j}) \right| \tag{139}$$

Now we need to upper-bound $\left| \zeta_i(X) - \zeta_i(X_{\backslash j}) \right|$ for all the different $i$'s. The cases are as follows:

**Case I: $i = j$** . Since the upper bounds $|\zeta_i(X)| \leq (2d+1)\log N$ and $\left| \zeta_i(X_{\backslash j}) \right| \leq (2d+1)\log(N-1)$ always holds, thus we have $\left| \zeta_i(X) - \zeta_i(X_{\backslash j}) \right| \leq 2(2d+1)\log N$. Since there's only one $i$ equal to $j$, we have $\sum_{\text{Case I}} \left| \zeta_i(X) - \zeta_i(X_{\backslash j}) \right| \leq 2(2d+1)\log N$.

**Case II: $i \neq j$ and $\rho_{k,i} = 0$** . Suppose $S \in \{\mathrm{pa}(X_l)\}_{l=1}^{d} \cup \{\mathrm{pa+}(X_l)\}_{l=1}^{d} \cup \{X\}$ is a subset of $X$. Since $\rho_{k,i} = 0$, then $n_S^{(i)} = \left| \{i' \neq i : s^{(i)} = s^{(i')}\} \right|$. We recall that for $S = X$ we actually have $n_S^{(i)} = \tilde{k}_i$. Thus by removing the point $x^{(j)}$ is this case, for any subset $S$, if $s^{(i)} = s^{(j)}$, then $n_S^{(i)}$ is

decreased by 1, and if $s^{(i)} \neq s^{(j)}$ then $n_S^{(i)}$ will not change. Thus we can write:

$$\left| \zeta_i(X) - \zeta_i(X_{\setminus j}) \right| \tag{140}$$

$$\leq \left| \psi\left(\tilde{k}_i\right) - \psi\left(\tilde{k}_i - 1\right) \right| \tag{141}$$

$$+ \sum_{l=1}^{d} \mathbf{1}_{\{\mathrm{pa}(X_l) \neq \emptyset\}} \mathbf{1}_{\left\{x_{\mathrm{pa}(l)}^{(i)} = x_{\mathrm{pa}(l)}^{(j)}\right\}} \left| \log(n_{\mathrm{pa}(X_l)}^{(i)} + 1) - \log n_{\mathrm{pa}(X_l)}^{(i)} \right| \tag{142}$$

$$+ \sum_{l=1}^{d} \mathbf{1}_{\left\{x_{\mathrm{pa+}(l)}^{(i)} = x_{\mathrm{pa+}(l)}^{(j)}\right\}} \left| \log(n_{\mathrm{pa+}(X_l)}^{(i)} + 1) - \log n_{\mathrm{pa+}(X_l)}^{(i)} \right| \tag{143}$$

$$+ \left( \sum_{l=1}^{d} \mathbf{1}_{\{\mathrm{pa}(X_l) = \emptyset\}} + 1 \right) \left( \log N - \log(N-1) \right) \tag{144}$$

$$\leq \frac{1}{\tilde{k}_i - 1} + \sum_{l=1}^{d} \mathbf{1}_{\{\mathrm{pa}(X_l) \neq \emptyset\}} \mathbf{1}_{\left\{x_{\mathrm{pa}(l)}^{(i)} = x_{\mathrm{pa}(l)}^{(j)}\right\}} \frac{1}{n_{\mathrm{pa}(X_l)}^{(i)}} \tag{145}$$

$$+ \sum_{l=1}^{d} \mathbf{1}_{\left\{x_{\mathrm{pa+}(l)}^{(i)} = x_{\mathrm{pa+}(l)}^{(j)}\right\}} \frac{1}{n_{\mathrm{pa+}(X_l)}^{(i)}} \tag{146}$$

$$+ \left( \sum_{l=1}^{d} \mathbf{1}_{\{\mathrm{pa}(X_l) = \emptyset\}} + 1 \right) \frac{1}{N-1} \tag{147}$$

Where we used the fact that $\log(1 + 1/x) \leq 1/x$. For any $S$, the number of nodes $i$ satisfying $s^{(i)} = s^{(j)}$ is no more than $n_S^{(j)}$, and the total number of points in the case II is no more than $N - 1$. Thus we can write:

$$\sum_{i \in \text{Case II}} \left| \zeta_i(X) - \zeta_i(X_{\setminus j}) \right| \leq (\tilde{k}_i - 1) \frac{1}{\tilde{k}_i - 1} \tag{148}$$

$$+ \sum_{l=1}^{d} \mathbf{1}_{\{\mathrm{pa}(X_l) \neq \emptyset\}} n_{\mathrm{pa}(X_l)}^{(i)} \frac{1}{n_{\mathrm{pa}(X_l)}^{(i)}} \tag{149}$$

$$+ \sum_{l=1}^{d} n_{\mathrm{pa+}(X_l)}^{(i)} \frac{1}{n_{\mathrm{pa+}(X_l)}^{(i)}} \tag{150}$$

$$+ \left( \sum_{l=1}^{d} \mathbf{1}_{\{\mathrm{pa}(X_l) = \emptyset\}} + 1 \right) (N-1) \frac{1}{N-1} \tag{151}$$

$$= 2(d+1) \tag{152}$$

**Case III: $i \neq j$ and $\rho_{k,i} > 0$.** In this case, $\tilde{k}_i$ is always equal to $k$, and for any subset $S \in \{\mathrm{pa}(X_l)\}_{l=1}^{d} \cup \{\mathrm{pa+}(X_l)\}_{l=1}^{d} \cup \{X\}$ we have $n_S^{(i)} = \left| \{i' \neq i : \|s^{(i)} - s^{(j)}\| \leq \rho_{k,i}\} \right|$.

*Case III.1: $\|x^{(i)} - x^{(j)}\| \leq \rho_{k,i}$.* This means that the point $x_j$ is in the $k$ nearest neighbors of $x_i$ and by eliminating it, $\rho_{k,i}$ will change. So we don't know how $n_S^{(i)}$'s will change, thus we will use the upper-bound $\left| \zeta_i(X) - \zeta_i(X_{\setminus j}) \right| \leq 2(2d+1) \log N$. From the first part of the Lemma C.1 from [51], we can upper bound the number of such $i$'s by $\gamma_d k$, in which $\gamma_d$ is the minimum number of $d$-dimensional cones with the angle smaller than $\pi/6$ needed to cover the total space $\mathbb{R}^d$. Thus we have:

$$\sum_{\text{Case III.1}} \left| \zeta_i(X) - \zeta_i(X_{\setminus j}) \right| \leq 2k(2d+1)\gamma_{d_X} \log N \tag{153}$$

where $d_X$ is the dimension of the space $\mathcal{X}$.

*Case III.2:* $\|x^{(i)} - x^{(j)}\| > \rho_{k,i}$. This case is somewhat similar to the Case II. Here the $\rho_{k,i}$ will not change. But the $n_S^{(i)}$'s for all subsets $S \neq X$ can decrease by 1. We can write:

$$\sum_{\text{Case III.2}} \left| \zeta_i(X) - \zeta_i(X_{\setminus j}) \right| \tag{154}$$

$$\leq \sum_{i=1}^{N} \sum_{l=1}^{d} \mathbf{1}_{\{\text{pa}(X_l) \neq \emptyset\}} \mathbf{1}_{\left\{ x_{\text{pa}(l)}^{(i)} = x_{\text{pa}(l)}^{(j)} \right\}} \left| \log(n_{\text{pa}(X_l)}^{(i)} + 1) - \log n_{\text{pa}(X_l)}^{(i)} \right| \tag{155}$$

$$+ \sum_{i=1}^{N} \sum_{l=1}^{d} \mathbf{1}_{\left\{ x_{\text{pa+}(l)}^{(i)} = x_{\text{pa+}(l)}^{(j)} \right\}} \left| \log(n_{\text{pa+}(X_l)}^{(i)} + 1) - \log n_{\text{pa+}(X_l)}^{(i)} \right| \tag{156}$$

$$+ \sum_{i=1}^{N-1} \left( \sum_{l=1}^{d} \mathbf{1}_{\{\text{pa}(X_l)=\emptyset\}} + 1 \right) \left( \log N - \log(N-1) \right) \tag{157}$$

$$\leq \sum_{i=1}^{N} \sum_{l=1}^{d} \mathbf{1}_{\{\text{pa}(X_l) \neq \emptyset\}} \mathbf{1}_{\left\{ x_{\text{pa}(l)}^{(i)} = x_{\text{pa}(l)}^{(j)} \right\}} \frac{1}{n_{\text{pa}(X_l)}^{(i)}} \tag{158}$$

$$+ \sum_{i=1}^{N} \sum_{l=1}^{d} \mathbf{1}_{\left\{ x_{\text{pa+}(l)}^{(i)} = x_{\text{pa+}(l)}^{(j)} \right\}} \frac{1}{n_{\text{pa+}(X_l)}^{(i)}} \tag{159}$$

$$+ \sum_{i=1}^{N-1} \left( \sum_{l=1}^{d} \mathbf{1}_{\{\text{pa}(X_l)=\emptyset\}} + 1 \right) \frac{1}{N-1} \tag{160}$$

$$\leq \sum_{l=1}^{d} \mathbf{1}_{\{\text{pa}(X_l) \neq \emptyset\}} \gamma_{d_{\text{pa}(X_l)}} \log(N+1) \tag{161}$$

$$+ \sum_{l=1}^{d} \gamma_{d_{\text{pa+}(X_l)}} \log(N+1) + \sum_{l=1}^{d} \mathbf{1}_{\{\text{pa}(X_l)=\emptyset\}} + 1 \tag{162}$$

$$\leq 2d\gamma_{d_X} \log(N+1) + 1 \tag{163}$$

The inequality 162 follows from the second part of the Lemma C.1 from [51].

Now combining all three cases together, we have:

$$\sum_{i=1}^{N} \left| \zeta_i(X) - \zeta_i(X_{\setminus j}) \right|$$

$$\leq 2(2d+1)\log N + 2(d+1) + 2k(2d+1)\gamma_{d_X} \log N + 2d\gamma_{d_X} \log(N+1) + 1$$

$$\leq 6k(2d+1)\gamma_{d_X} \log(N+1) \tag{164}$$

Thus:

$$\sup_{x^{(1)},\ldots,x^{(N)},x'^{(j)}} \left| \widehat{\mathbb{GDM}}^{(N)}(X,\mathcal{G}) - \widehat{\mathbb{GDM}}^{(N)}(X^{(j)},\mathcal{G}) \right| \leq \frac{12k(2d+1)\gamma_{d_X} \log(N+1)}{N} \tag{165}$$

And:

$$\mathrm{Var}\left[\widehat{\mathbb{GDM}}^{(N)}(X,\mathcal{G})\right] \tag{166}$$

$$\leq \frac{1}{2}\sum_{j=1}^{N}\mathbb{E}\left[\left(\widehat{\mathbb{GDM}}^{(N)}(X,\mathcal{G}) - \widehat{\mathbb{GDM}}^{(N)}(X^{(j)},\mathcal{G})\right)^2\right] \tag{167}$$

$$\leq \frac{1}{2}\sum_{j=1}^{N}\sup_{x^{(1)},\dots,x^{(N)},x'^{(j)}}\left(\widehat{\mathbb{GDM}}^{(N)}(X,\mathcal{G}) - \widehat{\mathbb{GDM}}^{(N)}(X^{(j)},\mathcal{G})\right)^2 \tag{168}$$

$$\leq \frac{1}{2}\sum_{j=1}^{N}\left(\frac{12k(2d+1)\gamma_{d_X}\log(N+1)}{N}\right)^2 \tag{169}$$

$$= \frac{72\gamma_{d_X}^2\left(k(2d+1)\log N\right)^2}{N} \tag{170}$$

Therefore if $(k_N\log N)^2/N \to 0$ as $N$ goes to infinity, we have $\lim_{N\to\infty}\mathrm{Var}\left[\widehat{\mathbb{GDM}}^{(N)}(X,\mathcal{G})\right] = 0$.

## G   Details of Numerical Experiments

In this section, we will discuss the carried out numerical experiments done in more details. This includes but not limited to the choice of parameters, the data generated, and the derivation of the theory values. The parameter of the nearest neighbor $k$ for all the GDM, KSG and $\Sigma H$ methods is set to $\sqrt{N}/5$ to conform with the Assumptions 1 and keep the computational complexity at an acceptable level. The number of bins in binning method at each dimension is kept at $\sqrt[d]{N/m}$ for all the algorithms so we roughly have $m$ samples per each bin, and we choose $m \in \{10, 20, 100\}$ whichever giving the best precision.

### G.1   Experiment 1: Markov chain model with continuous-discrete mixture

For the first experiment, we simulated an $X$-$Z$-$Y$ Markov chain model in which the random variable $X$ is chosen as $X = \min\left(\alpha_1, \tilde{X}\right)$ where $\tilde{X} \sim \mathcal{U}(0,1)$ represents an auxiliary random variable uniformly distributed between 0 and 1. This means that we first generate a sample from $\tilde{X}$ denoted by $\tilde{x}$ and then let $x = \min\{\alpha_1, \tilde{x}\}$. Subsequently:

$$Z = \min(X, \alpha_2) \tag{171}$$
$$Y = \min(Z, \alpha_3) \tag{172}$$

We assume that $\alpha_3 < \alpha_2 < \alpha_1$. The three variables $X$, $Y$ and $Z$ represent a mixture of continuous and discrete random variables.

We simulated this system for various numbers of samples while setting $\alpha_1 = 0.9$, $\alpha_2 = 0.8$ and $\alpha_3 = 0.7$. For each set of samples $I(X;Y|Z)$ is estimated via different methods and its theory value is obviously equal to 0. The results are shown in Figure 2a.

### G.2   Experiment 2: Mixture of AWGN and BSC channels with variable error probability

As the second scheme of our experiments, we considered an Additive White Gaussian Noise (AWGN) Channel in parallel with a Binary Symmetric Channel (BSC) where only one of them can be activated at a time. The random variable $0 < Z < 1$ controls which channel is activated; i.e. if $Z$ is lower than the threshold $\beta$, then the AWGN channel is activated, otherwise the BSC channel will be activated.

The AWGN channel is modeled as $Y = X + N$ where $X \sim \mathcal{N}(0, \sigma_X^2)$ and $N \sim \mathcal{N}(0, \sigma_N^2)$. BSC channel is modeled as $Y = X \oplus E$, where $X$ and $E$ are two binary random variables $X \sim \mathrm{Bern}(p)$

and $E \sim \text{Bern}(Z)$ denoting the input and the error respectively. This means that the probability of error in the BSC channel is controlled by the variable $Z$ at each time-point. Equivalently, if we suppose that the sample $z_i$ is observed from $Z$ at the moment $i$, the output of the BSC at time $i$ is characterized by:

$$y_i = \begin{cases} x_i & \text{with probability } 1 - z_i \\ \neg x_i & \text{with probability } z_i \end{cases} \tag{173}$$

Let's assume $Z = \min\left(\alpha, \tilde{Z}\right)$ where $\tilde{Z} \sim U(0,1)$ is an auxiliary uniform random variable, similar to the previous experiments. The theory value for $I(X:Y|Z)$ is obtained as follows:

$$
\begin{aligned}
I(X;Y|Z) &= \int_{z=0}^{1} I(X;Y|Z=z) f_Z(z) dz & (174) \\
&= \int_{z=0}^{\beta} I_{\text{AWGN}}(X;Y|Z=z) f_Z(z) dz + \int_{z=\beta}^{1} I_{\text{BSC}}(X;Y|Z=z) f_Z(z) dz & (175) \\
&= \beta I_{\text{AWGN}}(X;Y) + \int_{z=\beta}^{\alpha} I_{\text{BSC}}(X;Y|Z=z) dz \\
&\quad + (1-\alpha) I_{\text{BSC}}(X;Y|Z=\alpha) & (176) \\
&= \frac{\beta}{2} \log\left(1 + \frac{\sigma_X^2}{\sigma_N^2}\right) + \int_{z=\beta}^{\alpha} \left(h(z\bar{p} + p\bar{z}) - h(z)\right) dz \\
&\quad + (1-\alpha)\left(h(\alpha\bar{p} + p\bar{\alpha}) - h(\alpha)\right) & (177)
\end{aligned}
$$

in which $h(x)$ is the binary entropy function defined as $h(x) = -x \log x - (1-x) \log(1-x)$ for $x \in [0,1]$.

In our experiment discussed in Section 5 we set $p = 0.5$, $\alpha = 0.3$, $\beta = 0.2$, $\sigma_X = 1$ and $\sigma_N = 0.1$. The theory value for the conditional mutual information can be readily calculated as $I(X;Y|Z) = 0.53241$. We simulated the system for various number of samples, and obtained the estimated CMI values $\hat{I}_N(X;Y|Z)$ for different methods. The results are shown in Figure 4a. Furthermore, we calculated the estimates of $I(X;Y|Z,Z^2,Z^3)$. Its theory value is obviously equal to $I(X;Y|Z)$, yet it's conditioned over a low-dimensional manifold in a high-dimensional space, and we are interested in examining the effect of it over the estimators' accuracy. The results are shown in Figure 4b.

## G.3 Experiment 3: Total Correlation for independent mixtures

In the third set of experiments, we estimated the total correlation of three independent random variables $X, Y$ and $Z$ each of which is created independently from a mixture distribution as follows: First we generate an auxiliary random variable $\tilde{X} \sim \text{Bern}(0.5)$ then the random variable $X$ is generated as follows:

$$X = \begin{cases} \alpha_X & \text{if } \tilde{X} = 0 \\ \sim \mathcal{U}(0,1) & \text{if } \tilde{X} = 1 \end{cases} \tag{178}$$

which means we toss a fair coin, if heads appears we will fix $X$ at $\alpha_X$, otherwise we will draw $X$ from a uniform distribution between $0$ and $1$. samples from $Y$ and $Z$ are also generated independently in the same fashion. For this setup, the theory value of the total correlation of the three variables $X$, $Y$ and $Z$ is obviously equal to $0$.

In our experiment we set $\alpha_X = 1$, $\alpha_Y = 1/2$ and $\alpha_Z = 1/4$, and generated various datasets with different number of samples. Then we estimated the total correlation via different approaches. The results are shown in the Figure 2c.

## G.4 Experiment 4: Total Correlation for independent uniforms with correlated zero-inflation

In this experiment we examine a system of random variables, which can be *clustered* into independent subsets, while inside each of the subsets, the variables are dependent. As a simple case we consider 4

random variables $X_1$, $X_2$, $X_3$ and $X_4$ which can be clustered as $\{X_1, X_2\}$ and $\{X_3, X_4\}$. Suppose $\tilde{X}_1$, $\tilde{X}_2$, $\tilde{X}_3$ and $\tilde{X}_4$ are four independent auxiliary random variables identically distributed as $\mathcal{U}(0.5, 1.5)$. Then we erase samples $(\tilde{X}_1, \tilde{X}_2)$ and $(\tilde{X}_3, \tilde{X}_4)$ independently, to generate the random variables $X_1$, $X_2$, $X_3$ and $X_4$; i.e. $X_1 = \alpha_1 \tilde{X}_1$, $X_2 = \alpha_1 \tilde{X}_2$ and $X_3 = \alpha_2 \tilde{X}_3$, $X_4 = \alpha_2 \tilde{X}_4$ in which $\alpha_1 \sim \text{Bern}(p_1)$ and $\alpha_2 \sim \text{Bern}(p_2)$. As we see, after this erasure process resulting in zero-inflation, $X_1$ and $X_2$ become correlated, so do $X_3$ and $X_4$ while still $(X_1, X_2) \perp\!\!\!\perp (X_3, X_4)$. Thus the total correlation can be written as:

$$
\begin{aligned}
TC\left(X_1, X_2, X_3, X_4\right) &= h(X_1) + h(X_2) + h(X_3) + h(X_4) - h\left(X_1, X_2, X_3, X_4\right) \qquad (179) \\
&= h(X_1) + h(X_2) + h(X_3) + h(X_4) - h\left(X_1, X_2\right) - h\left(X_3, X_4\right) (180) \\
&= I\left(X_1; X_2\right) + I\left(X_3; X_4\right) \qquad (181)
\end{aligned}
$$

To calculate $I\left(X_1; X_2\right)$, after applying the chain rule to $I\left(X_1; X_2, \alpha_1\right)$ twice, we have:

$$
\begin{aligned}
I\left(X_1; X_2\right) &= I\left(X_1; \alpha_1\right) + I\left(X_1; X_2 | \alpha_1\right) - I\left(X_1; \alpha_1 | X_2\right) \qquad (182) \\
&= h(\alpha_1) - \underbrace{h\left(\alpha_1 | X_1\right)}_{0} \qquad (183) \\
&\quad + p(\alpha_1 = 0) \underbrace{I\left(X_1; X_2 | \alpha_1 = 0\right)}_{0} + p(\alpha_1 = 1) \underbrace{I\left(X_1; X_2 | \alpha_1 = 1\right)}_{0} \\
&\quad - h\left(X_1 | X_2\right) + \underbrace{h\left(X_1 | X_2, \alpha_1\right)}_{=h(X_1|X_2)} \\
&= h(\alpha_1) \qquad (184)
\end{aligned}
$$

Similarly, $I\left(X_3; X_4\right) = h(\alpha_2)$. Thus:

$$
TC\left(X_1, X_2, X_3, X_4\right) = h(\alpha_1) + h(\alpha_2) \qquad (185)
$$

In the experiment, we set $p_1 = p_2 = 0.6$. The theory value for the total correlation is equal to 1.34602. The results of running different algorithms over the data can be seen in Figure 2d.

### G.5 Experiment 5: Gene Regulatory Networks

In this experiment we use different estimators to do Gene Regulatory Network inference based on the Restricted Directed Information (RDI) measure.

In a simplified model of a dynamical system $X = (X_1, \ldots, X_d)$, each $X_l$ is a time-series of length $T$ written in the form $\{X_l(t)\}_{t=0}^{T}$, and the system's evolution through time is characterized as $X_l(t) = g_l\left(X(t-1)\right) + N_l(t)$ in which $g_l(.) : \mathcal{X} \to \mathcal{X}_l$ is a deterministic function and $N_l(t)$ is an independent random noise. In the causal inference, the goal is to infer the set of $pa(X_l)$ for each $X_l$. To reach this end, many researchers have studied the information theoretic measures. One of such measures for time series is the directed information, and a variation of it is the restricted directed information defined as $RDI(X_i \to X_j) := I\left(X_i(t-1), X_j(t) | X_j(t-1)\right)$ and the conditional version of it (cRDI) is defined as $RDI(X_i \to X_j | Z) := I\left(X_i(t-1), X_j(t) | X_j(t-1), Z(t-1)\right)$. It's shown that for the first-order markov systems and under some mild conditions, $RDI(X_i \to X_j | \{X_i, X_j\}^c) \neq 0$ if and only if $X_i \in pa(X_j)$ [20]. Since RDI (cRDI) is in fact a conditional mutual information, its performance is directly related to that of the CMI estimator, which is used to obtain cRDI from the samples. Hence we are interested in evaluating the performance of various estimators in causal inference.

We do our test on the simulated neuron cells' development process, based on a model from [52]. In this model, the time series vector $X$ consists of 13 random variables each of which corresponding to a single gene in the development process. We simulated the development process for various values of $T$ in which the additive noise is independent and identically distributed as $\mathcal{N}(0, .03)$ for all the genes, and every single sample is then subject to erasure (i.e. be replaced by 0s) with a probability of 0.5. Then we applied the cRDI method to the data to discover the pairwise causal relationships. In our method, we first applied the RDI method (with no conditioning) and obtained the pairwise values. Then we repeated the process to obtain cRDI values, and for every pair $X_i$ and $X_j$, we conditioned the RDI over the $X_{k^*}$ in which $k^* = \text{argmax}_{k \neq i} RDI(X_k \to X_j)$.

Figure 4: The results for the experiments: 4a: The estimated CMI for the AWGN+BSC channels. 4b: CMI for the AWGN+BSC channels with low-dimensional $Z$ manifold. 4c: The AUROC values vs the number of samples for gene regulatory network inference with different estimators. The error bars show the standard deviations scaled down by $0.2$. 4d: The AUROC values vs the number of samples for feature selection accuracy with different estimators. The error bars show the standard deviations scaled down by $0.2$.

To calculate RDI and cRDI, various CMI estimators are utilized and then the Area-Under-ROC curve (AUROC) is calculated for each estimator. The more detailed results are shown in Figure 4c. We notice that the zero-inflation is highly detrimental to the causal signals, so we won't expect high performance of the causal inference over the data. As we see, RDI/cRDI methods implemented with the GDM estimator outperform the other estimators by at least %10 in terms of AUROC. We did not include the $\Sigma H$ estimator results due to its very low performance.

### G.6 Experiment 6: Feature Selection by Conditional Mutual Information Maximization

Selecting the best features for a learning task using information theoretic measures is well studied in the literature [7]. Among the well-known methods is the conditional mutual information maximization (CMIM) first introduced by Flueret [4], a variation of which was later introduced called CMIM-2 [53]. Suppose we have observed samples from the data $(X, Y)$, and would like to select the set $S \subset X$ which describes $Y$ the best. Suppose we start at $S = \emptyset$ and we add features to $S$ in a recursive greedy fashion as:

$$
\begin{aligned}
\text{If } S = \emptyset &: \quad S \leftarrow S \cup \{\text{argmax}_{X_i} I(X_i; Y)\} \\
\text{Otherwise} &: \quad S \leftarrow S \cup \{\text{argmax}_{X_i} f(X_i, S)\}
\end{aligned}
\tag{186}
$$

The algorithm 186 is equivalent to CMIM when $f(X_i, S) := \min_{X_j \in S} I(X_i, Y | X_j)$, and if we let $f(X_i, S) := \sum_{X_j} I(X_i, Y | X_j)$ we will get CMIM-2 instead.

In our experiment, we generated a vector $X = (X_1, \ldots, X_{15})$ of 15 random variables in which all the random variables are taken from $\mathcal{N}(0, 1)$ and then each random variable $X_i$ is clipped from above at $\alpha_i$ which is initially taken randomly from $\mathcal{U}(0.25, 0.3)$ and then kept constant during the sample generation. Then $Y$ is generated as $Y = \cos\left(\sum_{i=1}^{5} X_i\right)$. So only the first 5 features are relevant and the other features are independent of $Y$. Thus an ideal feature selection method should be able to recover the first 5 features first. We implemented and ran both CMIM and CMIM-2 algorithms with various CMI estimators to evaluate the performance of the estimators and see how well they can extract the true features $X_1, \ldots, X_5$. A more detailed version of AUROC plot including both CMIM and CMIM-2 is shown in Figure 4d . We can see that the feature selection methods implemented with the GDM estimator outperform the other estimators. We notice that the performances of CMIM and CMIM-2 with the GDM estimator are almost identical.