[Reviews · NeurIPS 2018]

Reviewer 1



This paper develops multivariate information measures in fairly general probability spaces based on Bayesian Networks. The references include a variety of references on multivariate and conditional information theory, although some related work has appeared at NIPS in recent years, such as [47] and other work by the authors of [47]. The reference list and prior art sections appear to be adequate, though explicit pointers to a set of papers prototypical of the \Sigma H paradigm would be useful. The extension to multivariate information measures is an original advance in the context of prior literature, and highlights some defects in prior literature (lines 180-183). The paper is well written overall. I feel like the technical content and style of this paper is more suited to a statistics or information theory venue, like IEEE Transactions on information theory, in line with many of the references presented, but is still within the purview of NIPS. Figure 3 is useful in the context of describing the proposed estimator. The presentation of numerical results in Figure 2 should include some confidence intervals for the estimates for better comparison of methods. The selection of experiments seems sufficient in Section 5, though calling your estimator GDM or something rather than "mixture" may make it a bit clearer. Update: Re-scored in light of author's responses and reviews.

Reviewer 2



This paper introduces consistent estimators for several information theoretic quantities, given access to iid samples from the underlying distribution. Known estimators impose very stringent assumptions on the underlying distribution (finite support or underlying density etc.), while the major thrust of this article is to introduce a method which is valid fairly generally, and has guarantees under minimal assumptions. I believe the problem to be of significant importance, and very well-aligned with the interests of this community. The paper is very well written, motivates the problem in very clear terms, and proposes a fairly general solution. The authors provide a very succinct summary of the state of the art, and compare their proposed procedure to other alternative and existing estimators through extensive numerical experiments. As a comment, I request the authors to also discuss situations where the Assumptions (L209) are violated, and scenarios where their estimator might be expected to behave poorly.

Reviewer 3



This paper considered the problem of estimating (multivariate) mutual information from the i.i.d. drawn data samples. While this problem has been rather extensively studied in the literature recently, most current methods focus on estimating entropy/differential entropy and then use them to build a plug-in estimator for mutual information. Apparently, this method does not work for general probability spaces with mixed value components. This paper proposed a new angle of viewing (multivariate) mutual information as divergences between a given distribution and a graphical model structure. The most important advantage of the proposed angle is that the Radon-Nikodym derivative is guaranteed to exist and thus the divergences are well defined for general probability spaces. The authors then proposed an estimator to estimate such divergences based on the coupling trick used to construct the well-known KSG estimator and proved the consistency of the proposed estimator. The paper is well organized and well written. In my opinion, the technical innovation is sufficiently novel for publication. My main complaint about this work is that just like the KSG estimator, the computational complexity of the proposed estimator is very high, especially for multi-dimensional distributions. This limits the applicability of the estimator in practice.